# Domain-inlaid Nme2Cas9 adenine base editors with improved activity and targeting scope

Nathan Bamidele [1], Han Zhang[1], Xiaolong Dong[2], Haoyang Cheng[1], Nicholas Gaston [1], Hailey Feinzig [1], Hanbing Cao[1], Karen Kelly[1], Jonathan K. Watts [1,3,4], Jun Xie [5,6,7,8], Guangping Gao [5,6,7,8] & Erik J. Sontheimer [1,8,9] ✉

Nme2Cas9 has been established as a genome editing platform with compact size, high accuracy, and broad targeting range, including single-AAV-deliverable adenine base editors. Here, we engineer Nme2Cas9 to further increase the activity and targeting scope of compact Nme2Cas9 base editors. We first use domain insertion to position the deaminase domain nearer the displaced DNA strand in the target-bound complex. These domain-inlaid Nme2Cas9 variants exhibit shifted editing windows and increased activity in comparison to the N-terminally fused Nme2-ABE. We next expand the editing scope by swapping the Nme2Cas9 PAM-interacting domain with that of SmuCas9, which we had previously defined as recognizing a single-cytidine PAM. We then use these enhancements to introduce therapeutically relevant edits in a variety of cell types. Finally, we validate domain-inlaid Nme2-ABEs for single-AAV delivery in vivo.

CRISPR-Cas9 DNA base editors (BEs) consist of a deaminase fused to a Cas9 nickase (nCas9) enzyme and enable precise nucleotide changes without double-strand breaks (DSBs)[1,2]. Upon single-guide RNA (sgRNA)-mediated target recognition, BEs deaminate specific bases within a defined editing window relative to the protospacer adjacent motif (PAM) in the displaced DNA strand. Currently, three classes of base editors exist: adenine BEs (ABEs), cytosine BEs (CBEs) and cytosine to guanine BEs (CBGEs), which enable A-to-G, C-to-T and C-to-G nucleotide conversion respectively[3–7]. With the ability to make precise nucleotide changes within the genome, BEs have the potential to address ~72% of disease-causing point mutations[1].

BE delivery remains a central challenge in therapeutic application in vivo, especially for extrahepatic tissues. Adeno-associated virus (AAV) vectors hold promise as an in vivo delivery approach for a variety of tissues but have a limited cargo capacity of ~4.7 kb. Some commonly used Cas9 orthologs (e.g., SpyCas9, which recognizes an NGG PAM) are large and incompatible with single AAV delivery. An avenue to address this issue is the use of smaller Cas9 orthologs. However, many of the initially identified compact Cas9s (e.g., Nme1-Cas9, SauCas9 and CjeCas9)[8–10], recognize 4-nucleotide protospacer adjacent motifs (PAMs), significantly reducing their targeting range and utility relative to those of Spy-BEs and their di-nucleotide PAM. Additional efforts have broadened BE targeting scope via engineered SpyCas9 variants with minimal PAM requirements (e.g., SpyCas9-NG, SpG, SpRY, and Spy-NRNH's)[11–13], although with undiminished delivery hurdles. In contrast, the development of compact Cas9 variants with

[1]RNA Therapeutics Institute, University of Massachusetts Chan Medical School, Worcester, MA 01605, USA. [2]Tessera Therapeutics, Somerville, MA 02143, USA. [3]Department of Biochemistry and Molecular Biotechnology, University of Massachusetts Chan Medical School, Worcester, MA 01605, USA. [4]Neuro-Nexus Institute, University of Massachusetts Chan Medical School, Worcester, MA 01605, USA. [5]Horae Gene Therapy Center, University of Massachusetts Chan Medical School, Worcester, MA 01605, USA. [6]Viral Vector Core, University of Massachusetts Chan Medical, School, Worcester, MA 01605, USA. [7]Department of Microbiology and Physiological Systems, University of Massachusetts Chan Medical School, Worcester, MA 01605, USA. [8]Li Weibo Institute for Rare Diseases Research, University of Massachusetts Chan Medical School, Worcester, MA 01605, USA. [9]Program in Molecular Medicine, University of Massachusetts Chan Medical School, Massachusetts, MA 01605, USA. ✉e-mail: erik.sontheimer@umassmed.edu

improved PAM ranges (e.g., SauCas9[KKH], SauriCas9, SchCas9)[14–16], potentially increase their utility as BEs, though these are still limited by di-nucleotide and purine-rich PAMs such as N3RRT, N2GG and NGR (N = any nucleotide and R = purine).

To expand the targeting scope and improve deliverability for base editing, we and others previously developed and characterized a compact ABE[17,18] composed of Nme2Cas9[19] and the laboratory-evolved tRNA adenosine deaminase, TadA8e[20]. Important properties of Nme2-ABE8e include an N4CC dinucleotide PAM that can target non-purine-containing sites, high sensitivity to target mismatches, and single -AAV (effector + guide) delivery capabilities. Although Nme2-ABE8e was effective in vivo following single-AAV delivery, editing levels were often inconsistent between target sites[17,18].

In this study, we have used structure-based[21] rational engineering approaches to further improve the editing activity and targeting scope of Nme2-ABE8e. We first used domain insertions[22–27] to re-position the deaminase (relative to the target strand) for improved base editing efficiency. As with SauCas9 and SpyCas9 BEs[22–27], we found that domain-inlaid Nme2-ABEs shift editing windows as well as improve editing efficiency. Domain-inlaid Nme2Cas9-deaminase fusions retained higher mismatch sensitivity than Spy-ABEs and were also compatible with CBEs as well as ABEs. We then expanded the targeting scope of the domain-inlaid Nme2-ABEs by exploiting our original identification of SmuCas9 and its single-cytidine (N4C) PAM requirement[28], as well as our demonstration that Nme1Cas9 and Nme2Cas9 can accommodate PAM-interacting domain (PID) swaps with other Cas9 homologs[19]. Specifically, we generated Nme2-BE derivatives with a transplanted SmuCas9 PID and validated them as effective BE platforms at single-C PAMs. Using the improved Nme2-ABE variants, we found that we can correct two common mutations that cause Rett syndrome [c.502 C > T (p.R168X) and c.916 C > T (p.R306C)] with little or no bystander editing and introduce additional therapeutically relevant edits by targeting splice sites for *DMD* and mouse *Cln3*. Lastly, we find that the domain-inlaid Nme2-ABE is highly active in vivo when delivered via single-AAV vector systems. The results improve the efficacy, targeting scope, and delivery capabilities of base editing systems in vivo.

## Results

### Development and characterization of domain-inlaid Nme2-ABEs
Structural analyses of Nme1Cas9 and Nme2Cas9[21], including the former in its cleavage-poised ternary complex, hinted that the inconsistent activity observed for the N-terminally fused Nme2Cas9 ABE variant (Nme2-ABE8e-nt) may be due to poor positioning of the deaminase domain relative to the predicted path of the displaced ssDNA target. We hypothesized that re-positioning of the deaminase closer to its target site may lead to increased editing efficiencies. A variety of Cas9 protein engineering approaches have been taken to alter the positioning of Cas9-domain fusions[29–31]. We opted for domain insertion, as several groups have shown that both Spy- and SauCas9 are amenable to this type of engineering[22–27]. Additionally, in the context of Cas9-BEs, the internal placement of a deaminase has been shown to decrease Cas9-independent off-target deamination, improving their safety profiles. We took a structure-guided approach[21] to select eight domain insertion sites at surface-exposed loops (Fig. 1a). Our initial panel of inlaid Nme2-ABE8e effectors (Nme2-ABE8e -i1 through -i8) include the TadA8e deaminase flanked by twenty amino acid (AA) flexible linkers inserted into the RuvC-inactivated Nme2Cas9 nickase mutant (nNme2[D16A]) (Supplementary Fig. 1a). To streamline the initial screening of the Nme2-ABE8e constructs, we used a previously described HEK293T ABE mCherry reporter cell line that is activated upon A-to-G conversion[17,32] (Fig. 1b). All domain-inlaid Nme2-ABE8e variants except Nme2-ABE8e-i4 activated the ABE reporter cell line above background levels, with several exhibiting efficiencies greater than that of the N-terminally fused version (Nme2-ABE8e-nt) (Fig. 1b).

Domain-inlaid base editors have been shown to shift the editing window depending on the site of deaminase insertion[24,26]. We reasoned that the mCherry reporter, with its single target adenosine at nt 8 of the 24-nt protospacer (A8, counting from the PAM-distal end), would not accurately reflect the editing activity of all Nme2-ABE8e effectors. To provide a more comprehensive view of editing windows, we analyzed the Nme2-ABE8e effectors at 15 endogenous target sites within HEK293T cells via plasmid transfections. Among the target sites tested, we found all domain-inlaid variants improved overall editing efficiencies ranging from a 1.05- to 2.27-fold increase compared to Nme2-ABE8e-nt (describing cumulative average editing for the 15 proto-spacers, Supplementary Fig. 1c). Encouraged by the activities of the domain-inlaid Nme2-ABE effectors, we explored how they compare to Spy-ABE8e at eight dual PAM targeting sites that have NGGNCC PAM regions (Fig. 1c–e). For this experiment, we focused on Nme2-ABE8e-i1, -i7, and -i8 because they exhibited the highest activities and most varied editing windows. The inlaid Nme2-ABE8e effectors showed comparable activity as Spy-ABE8e at six out of eight of the target sites (Fig. 1c, d). Furthermore, editing hotspots were altered in the inlaid versions in a manner consistent with the sites of deaminase insertion (Fig. 1e). Specifically, Nme2-ABE8e-i1 favored editing of PAM-distal adenosines, whereas the -i7 and -i8 effectors exhibited more PAM-proximal editing windows (Fig. 1e, Supplementary Fig. 1b). These results demonstrated that positioning of the deaminase relative to the targeted R-loop can improve the efficiency and alter the editing window of Nme2-ABEs. Because of their editing efficiencies and distinct editing windows, our subsequent analyses of inlaid Nme2-ABE8e variants focused primarily on the -i1, -i7 and -i8 effectors.

### Nme2Cas9 tolerates insertion of alternative deaminases
We next investigated whether Nme2Cas9 tolerates the insertion of cytosine deaminases (Nme2-CBEs). For these experiments, we again focused on the inlaid designs with insertion sites -i1, -i7, and -i8. We first turned our attention to the cytidine deaminase evoFERNY[33], which has a similar size as TadA8e (161AA and 166AA respectively). To construct the Nme2-evoFERNY effectors we used the same architecture as the domain-inlaid Nme2-ABEs, with the addition of a C-terminal 10AA linker and a single uracil glycosylase inhibitor (UGI) domain (Supplementary Fig. 2a). In addition to evoFERNY, we also constructed Nme2-CBEs with the larger rAPOBEC1 (rA1, 228AA) cytidine deaminase (Nme2-rA1)[3].

We tested the domain-inlaid Nme2-CBEs against their N-terminal fusion counterparts (Nme2-evoFERNY-nt or Nme2-rA1-nt), at three high-activity target sites in HEK293T cells by plasmid transfection. All Nme2-CBE effectors were functional at the genomic sites tested (Supplementary Fig. 2b). Like the domain-inlaid Nme2-ABEs, the Nme2-CBEs exhibited insertion-site-dependent shifts of editing hotspots for the target sites tested (Supplementary Fig. 2c). In addition, we noticed divergent editing patterns occurring at the same genomic loci between two analogous domain-inlaid Nme2-CBE effectors, some of which is likely attributed to the sequence specificity of the distinct cytidine deaminases (Supplementary Fig. 2b, 2c). These results demonstrate that Nme2Cas9 is a flexible scaffold for insertion of a variety of deaminase domains enabling C-to-T as well as A-to-G base editing.

### Chimeric Nme2[Smu]-ABEs enable recognition of N4CN PAMs, increasing their target scope
The activity of Cas9-BEs are limited to editing in specific editing windows specified by the distance from the PAM[2]. Here, we sought to increase the targeting scope of Nme2-ABEs by altering their PAM recognition properties. Our group previously demonstrated that PID swapping between closely related Type II-C Cas9 orthologs could alter their PAM preferences[19]. We also discovered and characterized Smu-Cas9 from *Simonsiella muelleri*, and found that it has a minimal PAM of N4CN, despite having limited nuclease editing activity in HEK293T cells[28]. We reasoned that chimeric, domain-inlaid Nme2-ABEs

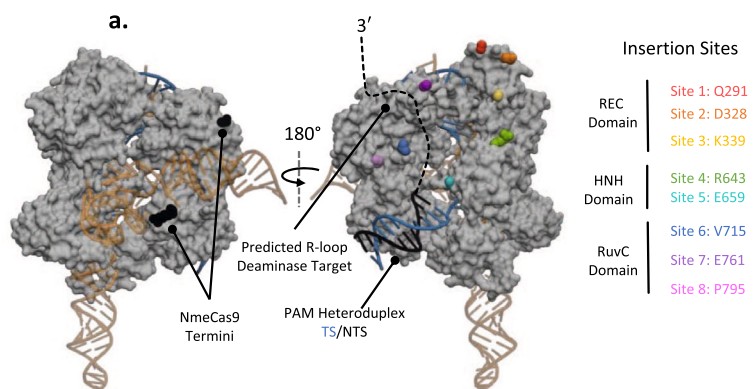

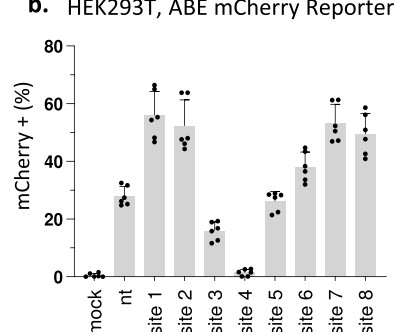

**b.** HEK293T, ABE mCherry Reporter

**c.** Editing Efficiency: HEK293T, Eight Dual-PAM Genomic Target Sites

**d.**

**e.** Editing Hotspots: HEK293T, Eight Dual-PAM Genomic Target Sites

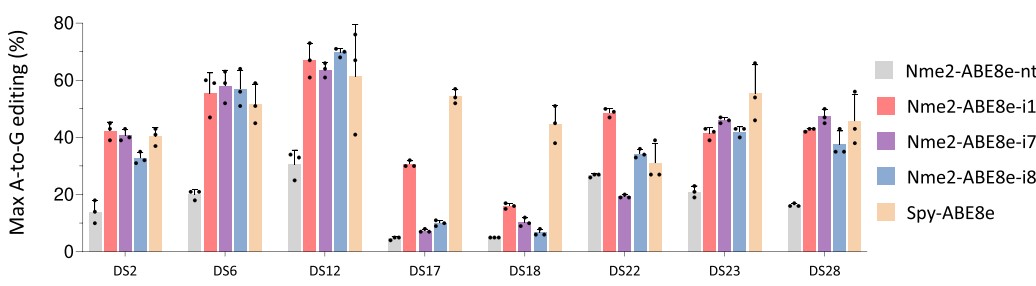

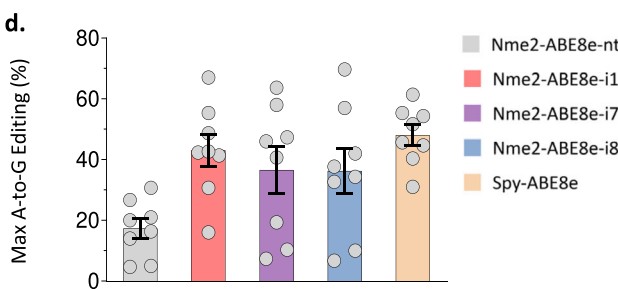

with an SmuCas9 PID (Nme2^Smu-ABEs) could alter the PAM preference from N4CC to N4CN, a four-fold increase in the number of target sites available for targeting by Nme2-ABEs.

After constructing the Nme2^Smu-ABE8e effectors with -i1, -i7 and -i8 designs (Fig. 2a), we tested their activities at a panel of five N4CC and nine N4CD (D = A, G or T) PAM targets in HEK293T cells by plasmid transfection (Supplementary Fig. 3a). For this experiment, Nme2-

ABE8e-i1 was used as a reference. For the N4CC targets, Nme2-ABE8e-i1 exhibited average editing of ~38% (describing the average maximally edited adenine across each protospacer in Supplementary Fig. 3a, 3b). All three Nme2^Smu-ABE8e effectors were also active at the N4CC target sites, but with 1.7- to 2-fold reductions in overall activity compared to Nme2-ABE8e-i1 (Supplementary Fig. 3b). As expected, Nme2-ABE8e-i1 had minimal activity at the N4CD target sites (Supplementary Fig. 3b).

**Fig. 1 | Design of domain-inlaid Nme2-ABEs. a** Nme1Cas9/sgRNA/DNA ternary complex structure, PDB:6JDV. Nme2Cas9 is 98% identical to Nme1Cas9 outside of the WED and PAM-interacting domains. Black spheres represent N- and C-termini and colored spheres represent sites of domain insertion. Deaminase domain insertion sites (Nme2Cas9 aa numbers) are specified to the right, with colors matching the sites indicated in the structure. **b** Activities of Nme2-ABE8e constructs in mCherry reporter cells (activated upon A-to-G editing) after plasmid transfection, measured by flow cytometry ($n = 3$ biological replicates in technical duplicate; data represent mean ± SD). **c** A-to-G editing following transfection of Spy-ABE8e vs. Nme2-ABE8e plasmids, using PAM-matched, endogenous HEK293T genomic loci. The editing efficiency at the maximally edited adenine for each target was plotted.

Editing efficiencies were measured by amplicon deep sequencing ($n = 3$ biological replicates; data represent mean ± SD). **d** Data from (**c**) were aggregated and replotted, with each data point representing the maximum A-to-G editing efficiency of an individual target site, as measured by amplicon deep sequencing ($n = 3$ biological replicates; data represent mean ± SEM). **e** Summary of mean A-to-G editing activities and editing windows for Spy- and Nme2-ABE8e constructs in HEK293T cells. Numbers provided for each position in the protospacer represent the mean A-to-G editing efficiency across eight PAM-matched endogenous target sites, as measured via amplicon deep sequencing ($n = 3$ biological replicates per target). Crossed-out boxes indicate that no adenine was present at the specified position in the target panel tested. Source data are provided as a Source Data file.

By contrast, all three PID-swapped, domain-inlaid Nme2$^{Smu}$-ABE effectors effectively installed A-to-G edits at the N$_4$CD PAM target sites, though with varied efficiencies (Supplementary Fig. 3b). These results indicate that Nme2$^{Smu}$-ABEs can target and install precise edits at sites with N$_4$CN PAMs.

### Editing windows and activities of Nme2- and Nme2$^{Smu}$- ABE variants

To further investigate the editing characteristics of the domain-inlaid Nme2- and Nme2$^{Smu}$-ABE8e effectors, we assessed their activities using a paired guide-target library approach[27,34–37]. The library consisted of 200 unique guide-target pairs cloned into a plasmid backbone flanked by Tol2 inverted terminal repeats enabling stable genome integration within HEK293T cells (Supplementary Fig. 4a; Supplementary Data 1, Oligonucleotides). Some guide-target pairs in the library corresponded to previously validated and analyzed sites[19,38,39], whereas other were included for their preclinical therapeutic development potential. Following integration, we tested the panel of editors by plasmid transfection and subsequently sequenced the libraries at an average depth of ~1800 per library member (Supplementary Fig. 4b). For this experiment, we also included the recently evolved eNme2-C ABE8e variant as it has a relaxed N$_4$CN PAM preference (akin to that of Nme2$^{Smu}$-ABE8e) as well as increased activity in comparison to Nme2-ABE8e-nt[38].

Consistent with results at endogenous HEK293T target sites, Nme2-ABE variants with a WT PID demonstrated robust activity at N$_4$CC PAM targets, with minimal activity on N$_4$CD target sites. In contrast, inlaid Nme2$^{Smu}$-ABE8e effectors demonstrated robust activities at N$_4$CN PAM target sites (Fig. 2b, c). For example, Nme2-ABE-i1 exhibited mean maximum editing efficiencies of ~15% at N$_4$CC PAM targets and ~4% at N$_4$CD PAM targets. By contrast, Nme2$^{Smu}$-ABE-i1 had efficiencies of ~19% at N$_4$CC PAM targets and ~29% at N$_4$CD PAM targets. The observed editing window for eNme2-C across all library members spanned positions 6-14 (referring to activity >50% of the window maximum), with a center of position 9, in agreement with eNme2-C's previously reported editing window and center[38] (Fig. 2d, Supplementary Fig. 5). Consistent with our endogenous HEK293T target site data, domain-inlaid Nme2- and Nme2$^{Smu}$-ABE8e effectors exhibited wide editing windows of 7-13 nucleotides and with a Tad8Ae insertion-site-dependent shift in editing window (Figs. 2c, d and Supplementary Fig. 5, 6). We next compared editing windows between WT and PID-swapped constructs at N$_4$CC PAM targets. Although window centers were identical between WT and PID-swapped effectors with the same insertion site, Nme2$^{Smu}$-ABE8e windows were smaller than those of Nme2-ABE8e's at N$_4$CC PAM targets (Supplementary Fig. 6). Observed window centers for ABE8e effectors with the -i1 insertion site fell between positions 7-8, whereas editing was centered around position 12 for -i8 effectors (Fig. 2c, Supplementary Figs. 5, 6).

### Analysis of domain-inlaid Nme2-ABE8e specificity

We then sought to determine the specificities of the domain-inlaid Nme2-ABEs. Guide-dependent off-target editing is driven by Cas9 unwinding and R-loop formation at targets with high sequence similarity[40]. We previously demonstrated that Nme2-ABE8e-nt has a much lower propensity for guide-dependent off-target editing compared to Spy-ABE8e[17]. Using the most active inlaid variant (Nme2-ABE8e-i1) as a prototype, we examined guide-dependent specificity using a series of double-mismatch guides targeting the mCherry reporter, with Spy-ABE8e and Nme2-ABE8e-nt used for comparison. In all cases, the target adenosine was at the eighth nt of the protospacer (Fig. 3a, b). To account for differences in on-target activity (especially for Nme2-ABE8e-nt), we normalized the activities of the mismatched guides to that of the respective non-mismatched guide. Consistent with our previous results, Spy-ABE8e significantly outperformed Nme2-ABE8e-nt for on-target activity (Fig. 3a), but exhibited far greater activity with mismatched guides (Fig. 3b). Nme2-ABE8e-i1 activated the reporter with a similar efficiency as Spy-ABE8e (Fig. 3a), but with greater sensitivity to mismatches (Fig. 3b). Although the Nme2-ABE8e-i1 variant was less promiscuous than Spy-ABE8e, it exhibited higher activity with mismatched guides than Nme2-ABE8e-nt, illustrating trade-offs between on- and off-target editing efficiencies observed previously elsewhere[40]. We then assayed the mismatch sensitivity of the Nme2-ABE8e -i7 and -i8 effectors, to determine whether their preference for PAM-proximal editing windows would alter the mismatch sensitivity in comparison to the -nt and -i1 effectors for activating the reporter cell line. In this experiment, Nme2-ABE8e-i7 and -i8 exhibited mismatch sensitivities comparable to Nme2-ABE-nt, while retaining high on-target activity (Fig. 3a, b). A potential explanation for the increased sensitivity of -i7 and -i8 effectors at this site is that the impact of imperfect base pairing between a guide and target may become more apparent as the optimal editing window shifts away from the target adenine. A recent strategy using imperfectly paired guide RNAs to minimize bystander editing relied on a similar concept, providing some support for this hypothesis[41].

Following the mismatch sensitivity assay, we evaluated the specificity of domain-inlaid Nme2- and Nme2$^{Smu}$-ABE8e's against their respective ABE8e-nt variants at bona fide endogenous off-target sites. Although Nme2Cas9 off-target sites are rare due to its intrinsic accuracy in mammalian genome editing[19], a few off-target sites have been identified for both nuclease and ABE variants via GUIDE-seq or in silico prediction. We selected four target sites for assessment, of which three had been validated as detectably edited off-target sites[17,19,38] (Fig. 3c). In agreement with the mismatch sensitivity assay, Nme2-ABE8e variants with domain insertion at the -i1 position exhibited the greatest increase in off-target editing efficiencies, reaching above 1% at two out of the four targets and yielding the least favorable specificity ratio [on-target:off-target editing ratio] of ~23:1. Also in agreement with the mismatch sensitivity assay, the -i7 and -i8 effectors displayed increased accuracy in comparison to the -nt effectors (with specificity ratios of ~200:1 for -i7, ~170:1 for -i8, and ~82:1 for -nt) (Fig. 3c).

Next, we turned our attention towards guide-independent off-target editing. We hypothesized that similar to other domain-inlaid BE architectures, the internal positioning of the deaminase would limit the propensity for off-target nucleic acid editing that occurs in trans. We used the orthogonal R-loop assay with HNH-nicking SauCas9 (nSau$^{D10A}$)[20,26,42] to generate off-target R-loops and capture the guide-

**a.**

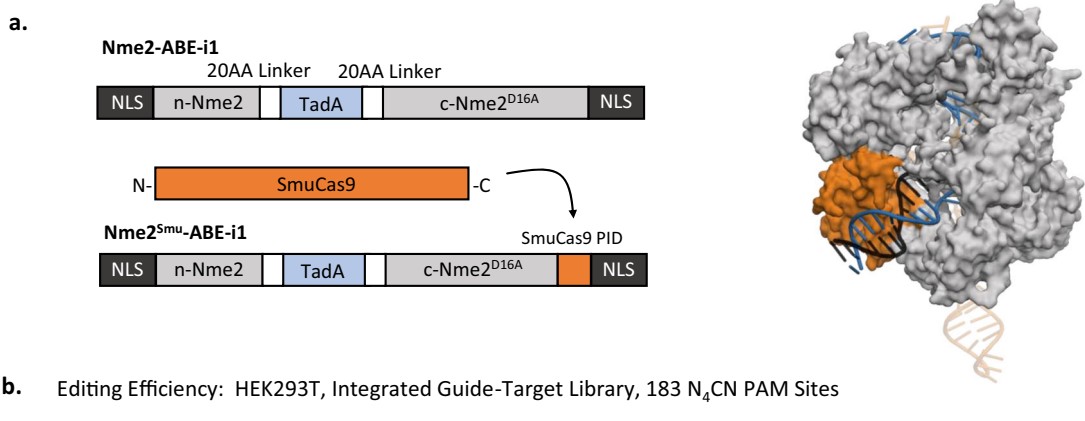

**b.** Editing Efficiency: HEK293T, Integrated Guide-Target Library, 183 N₄CN PAM Sites

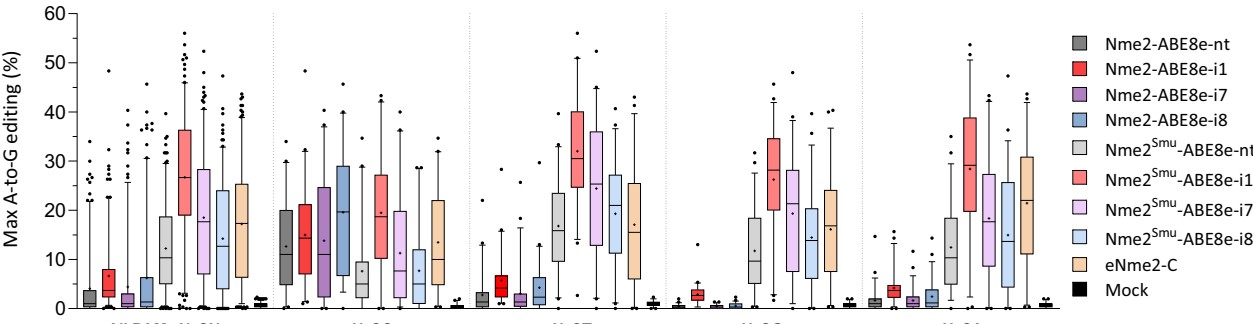

**c.** Editing Window: HEK293T, Integrated Guide-Target Library, 41 N₄CC PAM Sites

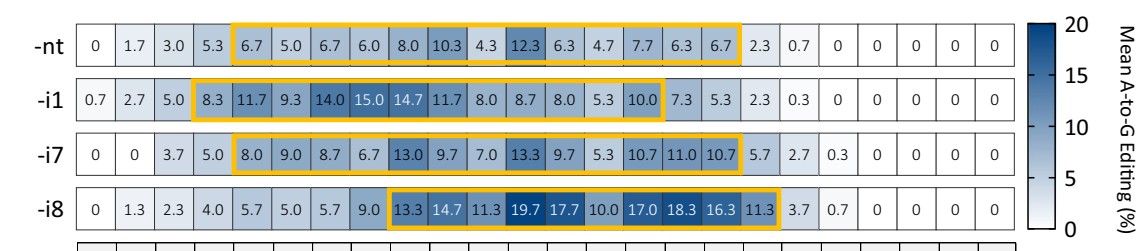

**d.** Editing Window: HEK293T, Integrated Guide-Target Library, 183 N₄CN PAM Sites

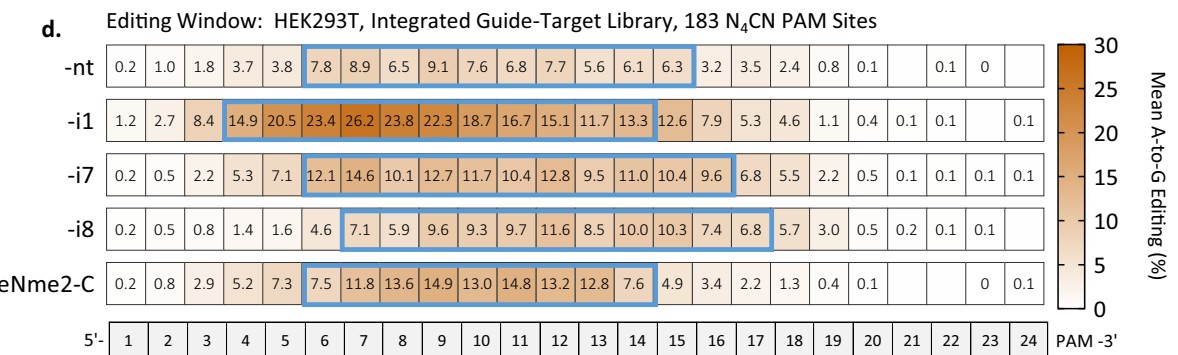

**Fig. 2 | PAM interacting domain (PID) chimeras expand the targeting scope of Nme2Cas9 effectors. a** Schematic of chimeric Nme2-ABE8e-i1 with the SmuCas9 PID (left). Homology model of Nme2$^{Smu}$Cas9 based on PDB:6JDV using the SWISS-MODEL program (right). **b** A-to-G editing following plasmid transfection of WT or chimeric PID Nme2-ABE8e effectors into HEK293T cells with 183 integrated paired guide-target sites with N₄CN PAMs. The editing efficiency at the maximally edited adenine for each target was plotted. Editing activities were measured by amplicon sequencing (*n* = 3 biological replicates; boxplots represent median and interquartile ranges; whiskers indicate 5th and 95th percentiles and the cross represents the mean). **c** Summary of mean A-to-G editing activities and editing windows for WT Nme2-ABE8e effectors at N₄CC PAM guide-target library members or (**d**) chimeric PID Nme2-ABE8e effectors and eNme2-C constructs at N₄CN PAM guide-target library members in HEK293T cells. Numbers provided for each position in the protospacer represent the mean A-to-G editing efficiency across the guide-target library members, as measured via amplicon deep sequencing (*n* = 3 biological replicates). Source data are provided as a Source Data file.

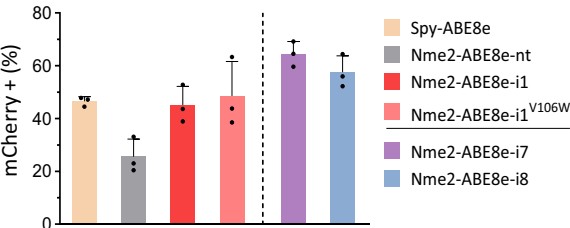

**a.** ABE Reporter: Activity of Perfectly Matched Guides

Legend:
- Spy-ABE8e
- Nme2-ABE8e-nt
- Nme2-ABE8e-i1
- Nme2-ABE8e-i1$^{V106W}$
- Nme2-ABE8e-i7
- Nme2-ABE8e-i8

**b.** ABE Reporter: Normalized Activity of Mismatched Guides

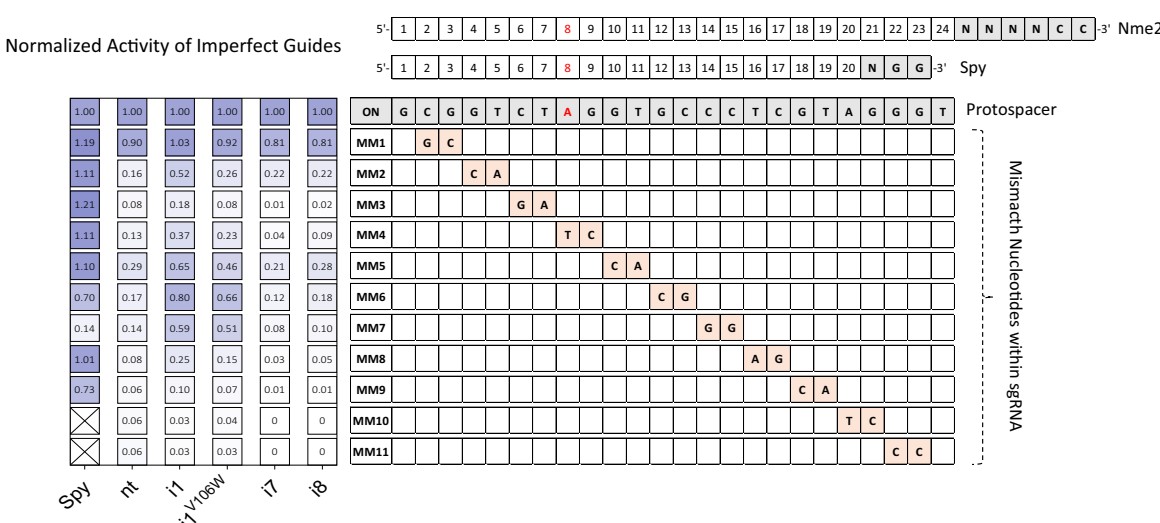

**c.** Editing Efficiency of Nme2-ABE8e Variants at Previously Validated Off-Target Sites

| Target Site | Mismatches Between On vs. Off | | Nme2-ABE8e | | | | Nme2$^{Smu}$-ABE8e | | | | mock |
|---|---|---|---|---|---|---|---|---|---|---|---|
| | | | nt | i1 | i7 | i8 | nt | i1 | i7 | i8 | |
| DS2 | GAATGGCAGGCGGAGGTTGTACTG**GGGGCC** | On- | 15.30 | 32.63 | 34.83 | 15.40 | 9.50 | 30.57 | 32.40 | 23.20 | 0.03 |
| | A--C--A--C--C-C--CTC---A**GTACCC** | Off- | 0.10 | 0.10 | 0.10 | 0.10 | 0.10 | 0.10 | 0.10 | 0.10 | 0.10 |
| Rosa26 | TGAGGACCGCCCTGGGCCTGGGAG**AATCCC** | On- | 48.87 | 65.67 | 66.13 | 63.97 | 44.67 | 60.73 | 62.87 | 61.15 | 0 |
| | GA--A-A----A-----------**ACTCCC** | Off- | 0.90 | 15.60 | 0.37 | 1.33 | 9.67 | 24.77 | 15.07 | 28.13 | 0 |
| otSG1 | TGCAGATCCCACAGGCGCCCTGGC**CAGTCG** | On- | 0.20 | 0.43 | 0.57 | 0.33 | 4.17 | 10.60 | 15.50 | 17.33 | 0 |
| | ----TT----------------**GATGCC** | Off- | 2.47 | 18.77 | 1.40 | 7.60 | 0.63 | 2.67 | 0.33 | 3.07 | 0.03 |
| otSG2 | AGTCTCCGCTTTAACCCCCACCTC**CAGCCG** | On- | 0.27 | 2.30 | 1.03 | 0.90 | 26.07 | 49.17 | 57.60 | 62.07 | 0 |
| | TTCTG-TTTAACCC--A--T---A**TCATAC** | Off- | 0.10 | 0.10 | 0.07 | 0.10 | 0.10 | 0.60 | 0.10 | 0.10 | 0.10 |

Max A-to-G editing observed at target (%)

independent DNA editing mediated by Spy-ABE8e or the Nme2-ABE8e variants (-nt and i1). We evaluated the on- and off-target activity of these ABE8e effectors by amplicon deep sequencing at the guide-targeted genomic site in addition to three SauCas9$^{D10A}$-generated R-loops. We found that Nme2-ABE8e-i1 was less prone to editing the orthogonal R-loops compared to Nme2-ABE8e-nt and Spy-ABE8e (Supplementary Fig. 7a). To account for differences in on-target

activities of the effectors, we reanalyzed the data by assessing the on-target: off-target editing ratio of each effector. Since Nme2-ABE8e effectors (-nt and -i1) have wider editing windows than Spy-ABE8e, we took the average editing activities across the respective windows of each effector for this target (protospacer positions 1-17nt for Nme2-ABE8e and 3-9nt for Spy-ABE8e), enabling a better comparison between the effectors (Supplementary Fig. 7b). In all cases, Nme2-

**Fig. 3 | Specificities of domain-inlaid Nme2Cas9-ABE8e variants. a** Comparison of on-target activity of transfected Spy-ABE8e and Nme2-ABE8e effectors in activating the ABE mCherry reporter, as measured by flow cytometry ($n = 3$ biological replicates; data represent mean ± SD). **b** Mismatch tolerance of Spy- or Nme2-ABE8e variants in ABE mCherry reporter cells at an overlapping target site positioning the target adenine for reporter activation at A8. Activities with single-guide RNAs carrying mismatched nucleotides as indicated (MM#, orange) are normalized to those of the fully complementary guides (ON, gray) ($n = 3$ biological replicates) for each effector, as indicated in the columns to the left. Heatmap data by column represent the normalized mismatched tolerance of the tested effectors.

**c** Comparison of Nme2-ABE8e variants at previously validated genomic targets. A-to-G editing was measured following transfection with WT or chimeric, PID-swapped Nme2-ABE8e plasmids at endogenous HEK293T or mouse N2A genomic loci following transfection. The editing efficiencies at the maximally edited adenine for the On- or Off-target site for each effector were marked in the heatmaps. Off-target mismatches to the spacer are denoted with red nucleotides, whereas dashes correspond to a matched nucleotide. Editing activities were measured by amplicon sequencing ($n = 3$ biological replicates; data represent mean). Source data are provided as a Source Data file.

ABE8e-i1 significantly outperformed Nme2-ABE8e-nt and Spy-ABE8e for guide-independent specificity at the orthogonal R-loops tested (Supplementary Fig. 7c). For this assay, we also investigated whether the TadA8e$^{V106W}$ mutant further increases the guide-independent DNA specificity with the Nme2-ABE8e-i1 architecture (Nme2-ABE8e$^{V106w}$-i1). As observed previously with other effectors[20], we observed increased specificity at all orthogonal R-loops with Nme2-ABE8e$^{V106w}$-i1 compared to Nme2-ABE8e-i1, though the specificity increase was only significant for R-loop 3 (*SSH2*) (Supplementary Fig. 7c).

**Nme2-ABEs enable correction of common Rett syndrome alleles**
Having established several Nme2-ABE variants with varied editing windows and PAM preferences, we sought to demonstrate their use in a disease-relevant context. We previously showed that Nme2-ABE8e-nt can correct the second-most-common Rett syndrome mutation (c.502 C > T; p.R168X)[17]. The c.502 C > T mutation resides within a pyrimidine-rich region, where the target adenine is not accessible by well-established single-AAV-compatible ABEs (e.g. SauCas9, SauCas9-KKH, and SauriCas9). Although promising, these initial experiments revealed the incidence of bystander editing at an upstream adenine (A16), resulting in a missense mutation of unknown consequence (c.496 T > C; p.S166P).

With the increased activity and shifted editing windows of the domain-inlaid Nme2-ABE8e variants, we hypothesized that we could avoid bystander editing by selecting guide RNAs that shift its position outside the editing window (Fig. 4a). To test whether the domain-inlaid Nme2-ABEs could correct c.502 C > T while avoiding bystander editing, we electroporated mRNAs with synthetic guides into Rett patient-derived fibroblasts (PDFs) bearing the c.502 C > T allele. We first tested the editing activities of the various effectors with our previously validated guide[17], denoted 502.G8. Consistent with our previous results, 502.G8 and Nme2-ABE8e-nt effectively corrected the target adenine (~19% efficiency), but with substantial (~10%) bystander editing (Fig. 4b). The domain-inlaid ABE8e variants were also active with 502.G8, with the -i7 and -i8 effectors exhibiting even higher bystander editing (~40–50%), likely reflecting their shifted editing windows, along with potential bystander editing of the wildtype allele (Fig. 4b). We next turned our attention to an additional guide, 502.G6, which places the target and bystander adenine at positions A16 and A22 of the protospacer respectively. With 502.G6, Nme2-ABE8e-nt performed poorly, exhibiting an average on-target editing efficiency of ~4% (Fig. 4b), whereas the -i1 and -i8 variants were somewhat more efficient (~14% and ~11% respectively). Importantly, 502.G6-guided correction by the domain-inlaid Nme2-ABE8e effectors occurred with undetectable editing at the A22 bystander (Fig. 4b).

Although the domain-inlaid Nme2-ABE8e effectors enabled correction of the c.502 C > T mutant with undetectable bystander editing, it came at the consequence of lowered on-target activity when using 502.G6 compared to 502.G8. We thus turned to the Nme2$^{Smu}$-ABE8e variants which allowed targeting the c.502 C > T with an additional four sgRNAs bearing non-canonical N$_4$CN PAMs, two of which (502.G9, 502.G10) placed the target and bystander adenines in favorable positions. Amplicon sequencing of the Nme2$^{Smu}$-ABE8e edited Rett PDF cells revealed that all the editors were inefficient at installing edits with

sgRNA 502.G9 (Source data). Conversely, sgRNA 502.G10 corrected the mutation more efficiently (~18% and ~16% for the -i7 and -i8 effectors respectively) than 502.G6 while avoiding bystander editing (Fig. 4b). We also tested editing of Rett PDF cells bearing the c.916 C > T (p.R306C) missense mutation (Fig. 4c). Using the Nme2$^{Smu}$-ABE8e variants at an N$_4$CT PAM we were also able to induce correction of this mutation. The -nt and -i1 effectors had average on-target editing rates of ~20%, ~22% respectively with bystander editing below 1% (Fig. 4d). We also tested an additional guide 916.G3 with an N$_4$CC PAM for the Nme2-ABE effectors, and although the on-target editing rate (~17% efficiency) was somewhat comparable to the PID swapped -i1 inlaid ABE variant, bystander editing was more substantial at this site (~5% efficiency) (Fig. 4d).

**Installation of therapeutic edits via splice site disruption with Nme2- and Nme2$^{Smu}$-ABEs**
We next explored the installation of additional therapeutically relevant edits by splice site disruption. ABEs can mediate exon skipping by editing consensus splice donor (SDS) or acceptor (SAS) sites, enabling gene disruption or ORF alteration without the introduction of double-strand breaks. These approaches are particularly advantageous for ABEs with wide editing windows, as the issue of bystander editing is minimized due to their presence within the intron or skipped exon.

First, we tested the ability of Nme2$^{Smu}$-ABE8e variants and eNme2-C editors to edit the SDS of *DMD* exon 50, an approach enabling *DMD* Δexon 51 reading frame restoration and a potential therapeutic approach for ~8% of DMD patients (Supplementary Fig. 8a)[43]. In this experiment, the domain-inlaid Nme2$^{Smu}$-ABEs performed similarly to eNme2-C depending on the target site, reaching SDS editing rates between 40-45% in HEK293T cells following plasmid transfection (Supplementary Fig. 8b).

Next, we tested the activities of eNme2-C, Nme2- and Nme2$^{Smu}$-ABE8e variants at 12 sites in Neuro2A cells targeting either the SAS or SDS of mouse *Cln3* exon 5. Deletion or skipping of *Cln3* exon 5 has been demonstrated to ameliorate disease phenotypes in a validated *Cln3* Δex7/8 mouse model[44,45] (Supplementary Fig. 8c). We observed on-target editing up to 15%, with domain-inlaid variants outperforming eNme2-C with 6 out of the 7 guides that exhibited significant activity (Supplementary Fig. 8d).

**Domain-inlaid Nme2-ABE8e enables in vivo base editing with a single AAV vector**
We previously developed and optimized a compact AAV design that enables all-in-one delivery of Nme2-ABE8e-nt with a sgRNA for in vivo base editing[17]. At 4996 bp, the cassettes harboring the domain-inlaid Nme2-ABE8e variants and a guide RNA are also within the packaging limit of some single AAV vectors, allowing us to test whether they outperform Nme2-ABE8e-*nt* in an in vivo setting. For our in vivo experiments we designed AAV genomes containing Nme2-ABE8e-nt, Nme2-ABE8e-i1 or Nme2-ABE8e$^{V106w}$-i1 with an sgRNA targeting the *Rosa26* locus (Fig. 5a).

We conducted two in vivo editing experiments with 9-week-old mice. First, we focused on systemic [intravenous (i.v.)] injection and editing in the liver, whereas the second experiment tested editing in

**a.** Rett PDF Cells, Heterozygous, *MeCP2* (c.502 C>T, p.R168X)

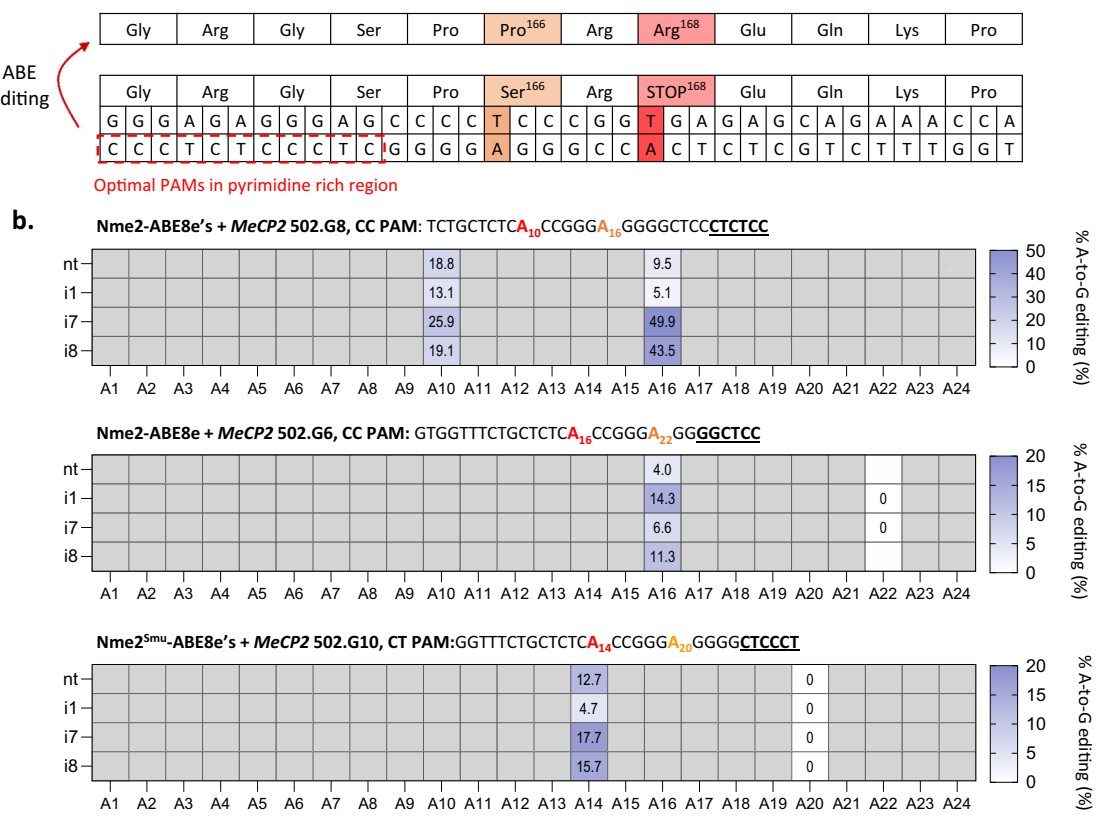

**b.** Nme2-ABE8e's + *MeCP2* 502.G8, CC PAM: TCTGCTCTCA₁₀CCGGGA₁₆GGGGCTCC**CTCTCC**

Nme2-ABE8e + *MeCP2* 502.G6, CC PAM: GTGGTTTCTGCTCTCA₁₆CCGGGA₂₂GG**GGCTCC**

Nme2^Smu-ABE8e's + *MeCP2* 502.G10, CT PAM: GGTTTCTGCTCTCA₁₄CCGGGA₂₀GGGG**CTCCCT**

**c.** Rett PDF Cells, Heterozygous, *MeCP2* (c.916 C>T, p.R306C)

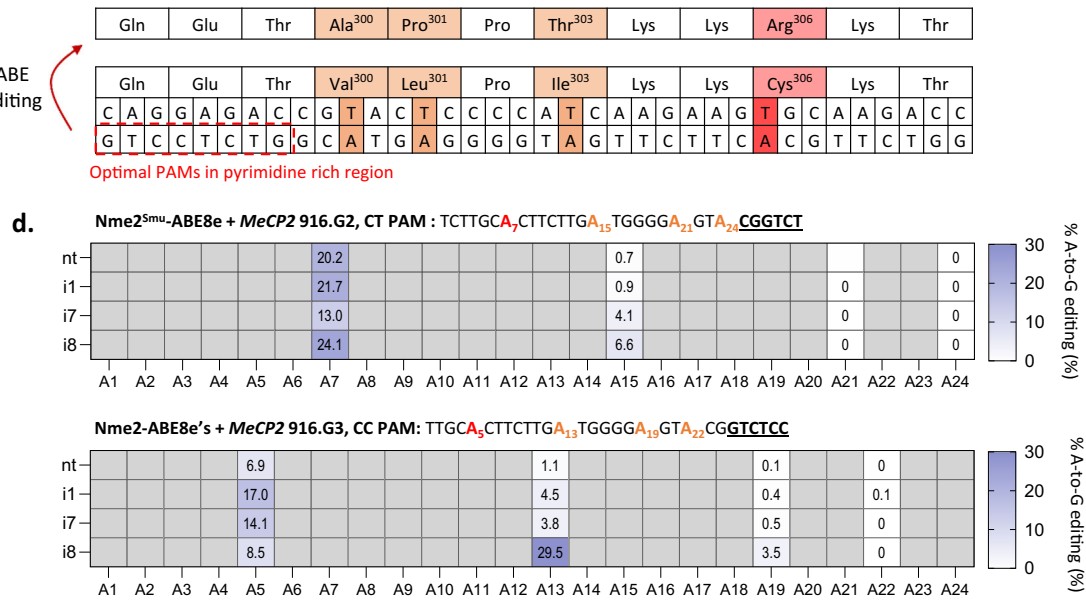

**d.** Nme2^Smu-ABE8e + *MeCP2* 916.G2, CT PAM : TCTTGCA₇CTTCTTGA₁₅TGGGGA₂₁GTA₂₄**CGGTCT**

Nme2-ABE8e's + *MeCP2* 916.G3, CC PAM: TTGCA₅CTTCTTGA₁₃TGGGGA₁₉GTA₂₂CG**GTCTCC**

**Fig. 4 | Correction of Rett Syndrome point mutations. a** Schematic of a portion of *MeCP2* exon 4, highlighting the (c.502 C > T; p.R168X) nonsense mutation in Rett patient-derived fibroblasts. **b** A-to-G editing of the *MeCP2* 502 C > T mutation in Rett patient fibroblasts in (**a**), measured by amplicon deep sequencing, with Nme2-ABE8e effectors delivered as mRNAs with synthetic sgRNAs (*n* = 3 biological replicates; data represent mean). Protospacer with target adenine (red), bystander adenine (orange), and PAM (bold, underlined). **c** Schematic of a portion of *MeCP2* exon 4, highlighting the (c.916 C > T; p. R306C) missense mutation in Rett patient-derived fibroblasts. **d** A-to-G editing of the *MeCP2* 916 C > T mutation in Rett patient fibroblasts in (C), measured by amplicon deep sequencing, with Nme2-ABE8e effectors delivered as mRNAs with synthetic sgRNAs (*n* = 3 biological replicates; data represent mean). Protospacers are shown with target adenine (red), bystander adenine (orange), and PAM (bold, underlined). Source data are provided as a Source Data file.

**a.**

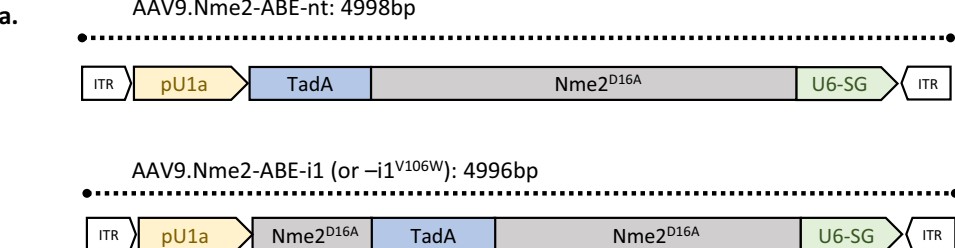

**b.**

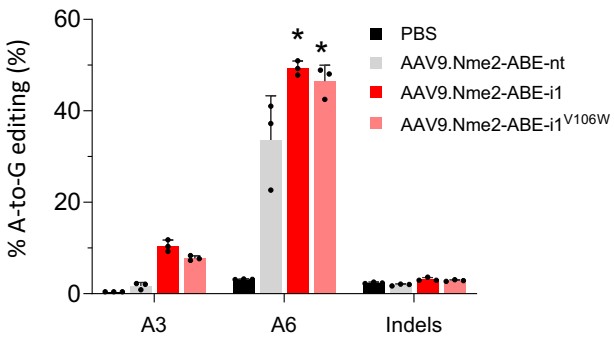
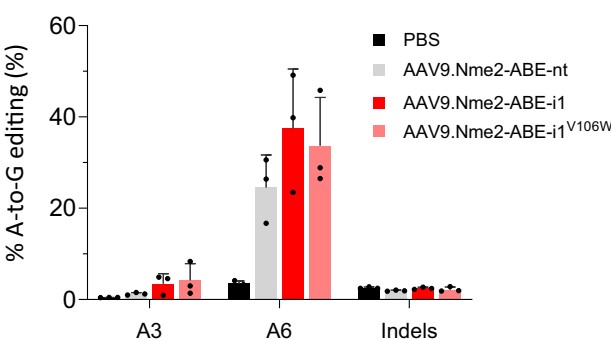

**c.**

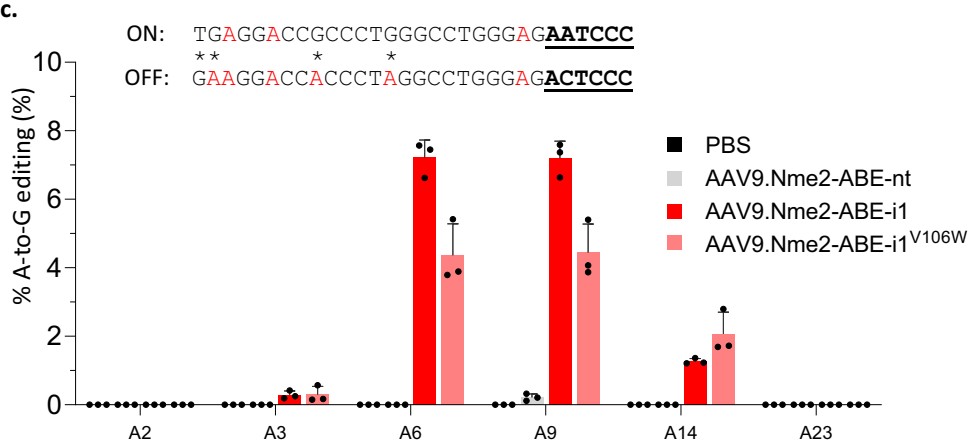

**Fig. 5 | In vivo editing with AAV9.Nme2-ABE8e-nt vs. –i1 vs. –i1^{V106W}. a** Schematic of the AAV constructs for the Nme2-ABE8e effectors. **b** Editing with AAV Nme2-ABE vectors in mouse liver (left) and striatum (right). Left, quantification of the editing efficiency at the *Rosa26* locus by amplicon deep sequencing using liver genomic DNA from mice that were tail-vein-injected with the indicated vector at $4 \times 10^{11}$ vg/ mouse (*n* = 3 mice per group; data represent mean ± SD). Nme2-ABE8e-i1 (p = 0.04), Nme2-ABE-i1^{V106W} (p = 0.015). Right, quantification of the editing efficiency at the *Rosa26* locus by amplicon deep sequencing using striatum genomic DNA from mice intrastriatally injected with the indicated vector at $1 \times 10^{10}$ vg/side (*n* = 3 mice per

group; data represent mean ± SD). One-way ANOVA analysis: ns, p > 0.05; *, p ≤ 0.05. **c** Protospacer of the *Rosa26* on-target site ("ON") and a previously validated Nme2-ABE8e off-target site (OT1, "OFF"). Adenines are in red, mismatches in OT1 have asterisks, and PAM regions are bold and underlined. The bar graph shows quantification of A-to-G edits in amplicon deep sequencing reads at the OT1 site using liver genomic DNA from mice tail-vein injected in (**b**), with vectors indicated in the inset (*n* = 3 mice per group; data represent mean ± SD). Source data are provided as a Source Data file.

the brain after intrastriatal injection. In both cases, mice were sacrificed 6 weeks after their respective injections and editing was quantified by amplicon sequencing. Within the liver, Nme2-ABE8e-i1 and Nme2-ABE-i1^{V106W} had editing efficiencies of ~49% (p = 0.015) and ~46% (p = 0.04) respectively, outperforming Nme2-ABE8e-nt (editing efficiency ~34% at A6 of the *Rosa26* target site), (one-way ANOVA) (Fig. 5b). Within the striatum the trend continued, with both Nme2-ABE8e-i1 and

Nme2-ABE-i1^{V106W} exhibiting improved editing activities (~37% and ~34% at A6 of *Rosa26*), compared to Nme2-ABE-nt (~25%), albeit this improvement did not reach statistical significance (p = 0.26 and 0.5, for Nme2-ABE8e-i1 and Nme2-ABE-i1^{V106W} respectively) (Fig. 5b).

We next sought to determine whether the boost in on-target activity in the liver was also accompanied by increased sgRNA-dependent off-target activity. The *Rosa26* sgRNA used in this study is

unusual among Nme2Cas9 guides in having a previously validated off-target site (*Rosa26*-OT1)[19]. We conducted amplicon sequencing at *Rosa26*-OT1 on genomic DNA extracted from the mouse livers used for our on-target analysis. We found that both Nme2-ABE8e-i1 and the V106W variant increased off-target A-to-G editing (up to ~7% and ~5% respectively) compared to Nme2-ABE8e-nt (~0.2%) (Fig. 5c). Collectively these results demonstrate that the increased activity of the domain-inlaid ABEs can translate to an in vivo setting, though this increase can come at the cost of increased off-target editing.

## Discussion

With the ability to enable single-nucleotide changes without genomic DSBs, BEs make exceptional candidates for use in the clinic as precision genome editors. This has been exemplified by the approval of clinical trials using BEs for therapeutic gene editing (clinicaltrials.gov identifiers: NCT05398029 and NCT05456880). Nonetheless, BEs face multiple hurdles to reach their full clinical potential. First, commonly used Spy-BEs are large, complicating delivery with AAV vectors, one of the most clinically advanced in vivo delivery vehicles for extrahepatic tissues. Second, the use of compact Cas9 orthologs compatible with single-vector AAV delivery are often limited by restrictive PAMs, limiting the scope of genomic sites that they can target. To this end we[17] and others[18] previously developed a compact ABE, Nme2-ABE-nt, enabling single-AAV delivery with a minimal $N_4CC$ dinucleotide PAM and high specificity. Nonetheless, despite the functionality of Nme2-ABE-nt in vivo at clinically relevant AAV doses, we observed inconsistent editing between target sites and significantly reduced editing activity in mammalian cells when compared to its Spy-ABE counterpart.

In this study, we used rational design to develop a panel of domain-inlaid Nme2-ABEs with increased editing activity, distinct editing windows, and a single-nucleotide PAM. Using the improved Nme2-ABE variants, we found that we can address common disease-causing point mutations such as certain *MeCP2* Rett syndrome alleles [(c.502 C > T; p.R168X) and (c.916 C > T, p.R306C)] with minimal bystander editing. We also found that the domain-inlaid variants are highly active in vivo. After systemic injection of Nme2-ABE8e-i1 AAV vector into adult mice at clinically relevant doses ($4×10^{11}$ vg/mouse), we observed nearly 50% editing in the liver. Although Nme2-ABE8e-i1 increased on-target activity within the liver compared to Nme2-ABE8e-nt, this was accompanied by increased off-target activity. Our specificity characterization of Nme2-ABE- i1, -i7 and -i8 demonstrated that in certain contexts they can achieve editing activities comparable to Spy-ABE while reducing guide-dependent and guide-independent off-target activity. Nevertheless, even with improved specificity compared with Spy-ABE, guide design and selection should be carefully considered to minimize undesired editing outcomes.

Our development of PID-chimeric Nme2$^{Smu}$-ABEs that recognize a single-cytidine PAM, further increases the utility of these compact BEs. While our work was in progress, others also demonstrated that the SmuCas9 PID can be used to create chimeric Cas9 nucleases with a $N_4CN$ PAM, even though their Nme2$^{Smu}$Cas9 nuclease exhibited modest efficiency at the target sites they assessed[39]. Although our Nme2$^{Smu}$-ABEs had detectable activity at all $N_4CN$ PAM target sites tested, we observed a reduction in activity at sites with $N_4CC$ PAMs when compared to domain-inlaid Nme2-ABE with a wildtype PID, a trend more pronounced at endogenous target sites. This result suggests that in cases where a target site contains a $N_4CC$ PAM, Nme2-ABE with a wildtype PID is the variant of choice. However, when single-cytidine PAMs are required, Nme2$^{Smu}$-ABEs can allow such sites to be targeted. Further protein engineering efforts, such as structure-guided design, mutational scanning, directed evolution, or a combination of all may be employed with Nme2$^{Smu}$-ABE or Nme2$^{Smu}$Cas9 as a starting scaffold to develop mutants with improved activity on all $N_4CN$ PAM targets, as has been demonstrated with eNme2-C[46].

For single-vector AAV delivery of the Nme2$^{Smu}$-ABE effectors, the slightly larger PID of SmuCas9 adds 24 bp (8aa) to the lengths of Nme2-ABEs, slightly exceeding the AAV packaging limit in our design with the U1a (251 bp) promoter. Alternative promoters such as the EF-1α short (EFS, 212 bp) promoter can be employed[18], likely enabling efficient packaging of Nme2$^{Smu}$-ABE effectors into a single AAV vector for in vivo editing. Alternatively, further optimization of the linker lengths between the nNme2Cas9 and deaminase domain (including for the domain-inlaid versions) promises to further minimize the sizes of both Nme2- and Nme2$^{Smu}$-ABE8e constructs.

Although the focus of this study was on the development and characterization of improved Nme2-ABEs, we found that our domain-inlaid architectures were also compatible with multiple CBE deaminases, laying a foundation for expanding the use of Nme2-BEs with improved activity and specificity for C-to-T and potentially C-to-G base editing. Overall, in light of Nme2-ABE's compact size, minimal PAM, specificity, shifted editing windows, and compatibility with single-AAV delivery, we anticipate that these advances will enable additional therapeutic applications with improved safety, efficacy, tissue tropism, and targeting range.

## Methods
### Ethical Statement
All animal study protocols were approved by the Institutional Animal Care and Use Committee (IACUC) at UMass Chan Medical School.

### Molecular cloning
Nucleotide sequences of Nme2Cas9 and Nme2$^{Smu}$Cas9 base editors described in this manuscript are provided in Supplementary Note 1. Plasmids expressing Nme2-ABE variants were constructed by Gibson assembly using Addgene plasmid #122610 as a backbone containing the CMV promoter and N- and C-terminal BP-SV40 NLSs. To generate Nme2-ABE-nt, the open reading frame of the N-terminally fused Nme2-ABE[17] was PCR-amplified and cloned into the CMV backbone. The domain-inlaid Nme2-ABEs were constructed with two sequential assemblies: first, nNme2$^{D16A}$ was assembled into the CMV backbone, and second, a gene block encoding the TadA8e domain and linkers was assembled into the assigned insertion sites. The domain-inlaid CBE deaminases were cloned in similar fashion to the ABE constructs, with Addgene #122610 as a backbone containing the CMV promoter, terminal BP-SV40 NLSs and a single UGI domain, with gene blocks encoding the evoFERNY[33] or rAPOBEC1 (rA1)[3] deaminase. Nme2-evoFERNY-nt was constructed via Gibson assembly by replacing nSpy$^{D10A}$ (Addgene #122610) with nNme2$^{D16A}$ and removing one of the UGI domains. Nme2-rA1-nt was subsequently cloned by replacing the evoFERNY domain with rA1 using the Nme2-evoFERNY-nt plasmid. Nme2-ABE-i1$^{V106W}$ was cloned by site-directed mutagenesis (SDM), using NEB's KLD enzyme mix (NEB #M0554S) with the appropriate Nme2Cas9 effector plasmid as a template. The nSauCas9$^{D10A}$ plasmid used for the orthogonal R-loop assay was also cloned by SDM using CMV-dSauCas9 (Addgene #138162) as a template. U6-driven sgRNA plasmids for the various Cas effectors were cloned using pBluescript sgRNA expression plasmids (Addgene #122089, #122090, #122091 for SpyCas9, SauCas9 and Nme2Cas9 respectively). In brief, the sgRNA plasmids were digested with BfuAI, followed by Gibson assembly with ssDNA bridge oligos containing a spacer of interest (G/N23 for Nme2Cas9, G/N19 for SpyCas9 and G/N21 for SauCas9). Nme2$^{Smu}$-ABE variants were cloned by replacing the Nme2Cas9 PID with the SmuCas9 PID using a gene block and Gibson assembly. The single-vector AAV plasmids were cloned by replacing the Nme2-ABE effector from our previously described AAV-Nme2-ABE8e_V2 plasmid[17] with the described domain-inlaid variants.

## In vitro mRNA synthesis

mRNAs used in this manuscript were in vitro transcribed as previously described[17], using the Hiscribe T7 RNA synthesis kit (NEB #E2040S). In brief, 500 ng of linearized plasmid template was used for the reaction, with complete substitution of uridine to 1-methylpseudouridine and CleanCap AG analog (N-1081 and N-7113, TriLink Biotechnologies).

## Transient transfection

Mouse N2A (ATCC #CCL-131), HEK293T (ATCC #CRL-3216) cells and their reporter-transduced derivatives were cultured in Dulbecco's Modified Eagle's Medium (DMEM; Genesee Scientific #25-500) supplemented with 10% fetal bovine serum (FBS; Gibco #26140079). All cells were incubated at 37 °C with 5% $CO_2$. For plasmid transfections, cells were seeded in 96-well plates at ~15,000 cells per well and incubated overnight. The following day, cells were transfected with plasmid DNA using Lipofectamine 2000 (ThermoFisher #11668019) following the manufacturer's protocol. For editing the mCherry reporter and endogenous target sites, 100 ng of effector plasmid and 100 ng of sgRNA plasmid was transfected with 0.75 μl Lipofectamine 2000. For the orthogonal R-loop assay, 125 ng of each effector and each sgRNA was used with 0.75 μl Lipofectamine 2000. For editing experiments with amplicon sequencing analysis, genomic DNA was extracted from cells 72 h post-transfection with QuickExtract (Lucigen #QE0905) following the manufacturer's protocol.

## Electroporation

Rett syndrome PDFs were obtained from the Rett Syndrome Research Trust and cultured in Dulbecco's Modified Eagle's Medium (DMEM; Genesee Scientific #25-500) supplemented with 15% fetal bovine serum (Gibco #26140079) and 1x nonessential amino acids (Gibco #11140050). These cells were also incubated at 37 °C with 5% $CO_2$. PDF electroporation's were performed using the Neon Transfection System 10 μl kit (ThermoFisher #MPK1096) as previously described[17]. A total of 500 ng ABE mRNA and 100 pmol sgRNA were electroporated into ~50,000 PDF cells. 48 h post-electroporation, genomic DNA was extracted with QuickExtract (Lucigen #QE09050) for amplicon sequencing.

## Flow cytometry

In total, 72 h post-transfection, cells were trypsinized, collected, and washed with FACS buffer (chilled PBS and 3% fetal bovine serum). Cells were resuspended in 300 μl FACS buffer for flow cytometry analysis using the MACSQuant VYB system. 10,000 cells per sample were counted for analysis with Flowjo v10.

## Amplicon sequencing and data analysis

Amplicon sequencing, library preparation, and analysis were performed as previously described[17]. Briefly, Q5 High-Fidelity polymerase (NEB #M0492) was used to amplify genomic DNA for library preparation, and libraries were pooled and purified twice after gel extraction with the Zymo gel extraction kit and DNA Clean and Concentrator (Zymo Research #11-301 and #11-303). Pooled amplicons were then sequenced on an Illumina MiniSeq system (300 cycles, Illumina sequencing kit #FC-420-1004) following the manufacturer's protocol. Sequencing data was analyzed with CRISPResso2[46] (version 2.0.40) in BE output batch mode with and the following flags: -w 12, -wc −12, -q 30.

## Guide-target library cloning

The 200-member guide-target library was designed and ordered as an oligo pool from Twist Bioscience (Supplementary Data 1. Oligonucleotides). The oligo pool was PCR-amplified according to the recommended Twist amplification protocol. The amplified pool was then cloned via Gibson assembly into p2Tol-U6-2xBbsI-sgRNA-HygR plasmid (Addgene, #71485) cut with XbaI and BbsI. The assembled

product was column-purified and electroporated into 10-beta electrocompetent cells (NEB #C3020K) as previously described[12,36] with the following adaptations. Following electroporation, the plasmid library was grown in an overnight liquid culture and isolated by miniprep plasmid purification. The number of transformants was assessed by serial dilution and counted colonies were above 200,000 for >1,000× library coverage.

## Guide-target library cell line generation and editing

Stable integration of the Tol2 guide-target library was achieved as previously described[36] with the following alterations. ~6 × 10^6 HEK293T cells in a 10-cm plate were transfected with 30 μg plasmid DNA at a 1:1 molar ratio of Tol2 transposase plasmid to guide-target plasmid library using Lipofectamine 2000 (ThermoFisher #11668019) and following the manufacturer's protocol. 1 day post-transfection, culture media was supplemented with hygromycin [50 μg ml⁻¹] for a minimum of 2 weeks before use in editing experiments. Library cells were maintained with over 200,000 cells for >1000× library coverage. The library cell line was transfected with ABE8e constructs that had been cloned into p2T-CMV-ABEmax-BlastR (Addgene, #152989) via Gibson assembly. For the transfections, cells were seeded with non-selective medium in 12-well plates at ~200,000 cells per well and incubated overnight. The following day, cells were transfected with 1.6 μg of plasmid DNA using Lipofectamine 2000 (ThermoFisher #11668019) following the manufacturer's protocol. 1 day post-transfection, culture media was supplemented with Blasticidin S [10 μg ml⁻¹]. After 3 days, genomic DNA was extracted from cells with QuickExtract (Lucigen #QE0905), column-purified and used for NGS library preparation.

## Guide-target library editing and analysis

NGS preparation and sequencing was done as described above with the following modifications. >1 μg of input DNA was used to ensure >500× library coverage[34], pooled amplicons were sequenced on an Illumina NextSeq 2000 system (200 cycles, Illumina sequencing kit #20046812) following the manufacturer's protocol. Sequencing data were further processed and binned by matching spacers and their barcode sequences using a custom demultiplexing script. Sequencing data was analyzed with CRISPResso2 (version 2.0.40) in BE output batch mode as described above. Guide-target library members with <40 reads were omitted from analysis in all samples.

## AAV production

AAV vector packaging was done at the Viral Vector Core of the Horae Gene Therapy Center at the UMass Chan Medical School as previously described[17]. Constructs were packaged in AAV9 capsids and viral titers were determined by digital droplet PCR and gel electrophoresis followed by silver staining.

## Mouse tail vein injection

All animal study protocols were approved by the Institutional Animal Care and Use Committee (IACUC) at UMass Chan Medical School. The 8-week-old C57BL/6 J mice (Jackson Laboratory, Stock No. 000664) were tail-vein injected with a dosage of $4 \times 10^{11}$ vg per mouse (in 200 μl saline). Mice were euthanized at 6 weeks post injection and perfused with PBS. Livers were harvested and pulverized in liquid nitrogen, and 15 mg of the tissue from each mouse liver was used for genomic DNA extraction. Genomic DNA from mouse liver or striatum (see below) was extracted using GenElute Mammalian Genomic DNA Miniprep Kit (Millipore Sigma #G1N350). Three mice per group were used to determine in vivo editing efficiency.

## Stereotactic intrastriatal injection

8-15-week-old C57BL/6 J mice were weighed and anesthetized by intraperitoneal injection of a 0.1 mg/kg Fentanyl, 5 mg/kg Midazolam,

and 0.25 mg/kg Dexmedetomidine mixture. Once pedal reflex ceased, mice were shaved and a total dose of $1 \times 10^{10}$ vg of AAV was administered via bilateral intrastriatal injection ($2\,\mu$l per side) performed at the following coordinates from bregma: +1.0 mm anterior-posterior (AP), ±2.0 mm mediolateral, and −3.0 mm dorsoventral. Once the injection was completed, mice were intraperitoneally injected with 0.5 mg/kg Flumazenil and 5.0 mg/kg Atipamezole and subcutaneously injected with 0.3 mg/kg Buprenorphine. Mice were euthanized at 6 weeks post-injection and perfused with PBS. Brains were harvested and biopsies at the striatum were taken for genomic DNA extraction.

## Western blot

Plasmids encoding C-terminal 6X-His tagged Nme2-ABE8e's were delivered with sgRNA into HEK293T cells via transient transfection as described above. Protein lysates were collected 72 h post-transfection by direct addition of 2x Laemmli sample buffer (BioRad #1610737EDU) followed by lysis at 95 °C for 10 min. Western blots were performed as described previously[47]. Primary mouse-anti-6xHis (ThermoFisher #MA1-21315, 1:2000 dilution) was used for Nme2-ABE8e detection and rabbit-anti-LaminB1 (Abcam #AB16048, 1:10,000 dilution) was used for detection of the loading control. After incubation with secondary antibodies, goat-anti-mouse IRDye®800CW (LI-COR #925-32210, 1:20,000 dilution) and goat-anti-rabbit IRDye®680RD (LI-COR #926-68071, 1:20,000 dilution), blots were visualized using a BioRad imaging system.

## Statistical analysis

Statistical analysis was performed using one- or two-way ANOVA using Dunnett's multiple comparisons test for correction in GraphPad Prism 9.4.0.

## Reporting summary

Further information on research design is available in the Nature Portfolio Reporting Summary linked to this article.

## Data availability

Sequencing data that support the findings of this study are available in the NCBI SRA bioproject PRJNA1033663. Source data for all figures are provided within the Source Data file. Sequences of target sites and oligonucleotides (primers, guide-target library oligos) used in this study are provided in the Supplementary Data 1. Oligonucleotides file. Plasmids described in this paper will be made available from Addgene. All other data are available upon request from the corresponding author. Source data are provided with this paper.

## Code availability

The custom demultiplexing script and associated demultiplexing reference sheet for the guide-target library analysis are available at https://github.com/SontheimerLab/guide-target-lib-demux[48]. All subsequent analysis was performed using publicly available programs with parameters indicated in the Methods section.

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

## Acknowledgements

We thank members of the Sontheimer, Gao, Xue, Watts, and Wolfe labs for their time, advice, and feedback during the conception of this manuscript. We also are grateful to the UMCMS Viral Vector Core for AAV packaging services and expertise, and to the Rett Syndrome Research Trust for patient-derived fibroblasts. Support for this work was provided by the National Institutes of Health (F31GM143879 and R25GM113686 to N.B. and R01GM150273 to E.J.S.), and by funding from the Leducq Foundation to E.J.S. and from the Rett Syndrome Research Trust to E.J.S. and J.K.W.

## Author contributions

N.B., X.D. and E.J.S. conceived the study and designed experiments with input from all authors. N.B., X.D., N.G., H.F. and H.A.C. performed and analyzed cell culture experiments. H.Z. performed and analyzed in vivo experiments using AAV vectors generated by J.X. K.K. performed mouse injections. H.C. developed and implemented scripts for computational analysis of sequencing data. J.K.W., G.G. and E.J.S. oversaw the experiments and analyses. N.B. and E.J.S. wrote the manuscript with editing contributions from all co-authors.

## Competing interests

The authors declare competing financial interests. The authors have filed patent applications on technologies related to this work. G.G. is a scientific co-founder of Voyager Therapeutics, Adrenas Therapeutics, and Aspa Therapeutics and holds equity in these companies. G.G. is an inventor on patents with potential royalties licensed to Voyager Therapeutics, Aspa Therapeutics, and other biopharmaceutical companies. E.J.S. is a co-founder and scientific advisor of Intellia Therapeutics and a scientific advisor of Tessera Therapeutics. The remaining authors declare that the research was conducted in the absence of any commercial or financial relationships that could be construed as a potential conflict of interest. The authors declare no competing non-financial interests.
