## [Peer Review File · Nature Communications]

Reviewers' Comments:

Reviewer #1:

Remarks to the Author:

In their manuscript, Bamidele et al. developed new base editors (BEs) combining the nickase version of Nme2Cas9 with different deaminases. The work built on previous work by the group in which they created an Nme2Cas9ABE8e variant that can be incorporated into Adeno-associated virus due to its compact size. They sought to improve the low activity of the variant by creating inlaid variants in which the deaminase domain was inserted at 8 different sites, based on the structure of Nme2Cas9, in the hope of better targeting of the deaminase domain to the non-targeted DNA strand. The 8 variants were tested on a genome-integrated cherry reporter system and the results were confirmed on 13 endogenous targets, where they could take into account the potentially different editing windows of the variants. Of the 3 most active inlaid variants, i1 showed a PAM distal, whereas i7 and i8 showed a PAM proximal editing window. Compared to SpCas9ABE8e, -i1 best approximated the activity of SpCas9ABE8e on 8 targets among the inserted variants.

Using the same insertion positions -i1, -i7 and -i8, two cytosine deaminases were integrated into Nme2Cas9 and their activity was monitored on 3 endogenous targets demonstrating that Nme2Cas9 is a flexible scaffold for insertion of a variety of deaminase domains.

In order to increase the targeting range, the PID (PAM interacting domain) of SmuCas9 which requires only a single 'C' PAM was replaced by the PID of Nme2 and a change in PAM preference was detected on 14 targets (5 on NNNNCC, 9 on NNNNCD PAM). Replacing the PID enabled the inlaid variants to function on NNNNCN targets.

The authors then tested the guide-dependent specificity of the variants with mismatch-containing guides in the cherry reporter system and at 4 genomic off-target sites. The inlaid variants showed higher off-target activity than the N-terminal variants, consistent with their higher on-target activity. The leader-independent off-target of the -i1, -i1-TadA8e V106W and N-terminal variants was investigated on SaCas9-generated R-loops. The inlaid variant showed reduced off-target activity compared to the N-terminal, while the TadA8e V106W mutation in the context of the inlaid variant further reduced off-target editing.

The utility of the variants was demonstrated in a patient-derived fibroblast by converting the second most common Rett syndrome mutation to WT, located in a pyrimidine-rich region where the previously developed base editors cannot be used. In addition, the activity of the -i1 inlaid variant was demonstrated in vivo by AAV delivery and in a mouse model.

This paper addresses an interesting problem by creating base editors that, due to their small size, can be packaged in AAV and applied to pyrimidine-rich regions. The manuscript is well structured, scientifically well written, the experiments are logical and fit for purpose, and the figures are descriptive. In general, the results of the experiments provide a good basis for the conclusions, and the interpretations are not exaggerated. I really have only minor comments:

-Supp. Fig. 2 and Supp. Fig. 3 are reversed, they are called the other way around in the text. In what follows I use the numbers assigned to the figures/figure captions.

-The legend of Supp. Fig. 2b mentions "On-target....R-loop assay". This term is not used in the MS text.

-In lane 155, the authors refer to Supplementary Fig. 2d. 'd' panel is not included in either Supp. Fig 2. nor Fig 3. Probably, it should be Supplementary Fig. 3c.

-Supp. Fig. 3a would be more informative if the figure also showed the position of UGI.

-Supp. Fig. 3b legend. Somehow it was inserted: "n=3 biological replicates per off-target R-loop" although there is no off-target experiment in the figure.

-Supp. Fig. 3c: The positions at the bottom of the figure run A1-A24. I assume they should run C1-C24.

-Supp. Fig. 3c legend:

"Summary of Mean C-to-T editing at the three endogenous HEK293T genomic loci with Nme2-evoFERNY or Nme2-rAPOBEC1 constructs." In contrast, the caption on the panel says that only the rAPOBEC1 result is shown, without the Nme2-evoFERNY one.

"Crossed out boxes denote no cytidine at the position within the target's tested." It appears that the positions shown here do not correspond to the target positions of 'C' shown in panel 'b', although the data are from the same experiments as I understood from the relevant text (lines 147-151).

-Fig 1e: When plotting the Dual-PAM spacer sequences, the two sequences are misaligned, as if

the PAM of SpCas9 falls on position 21-23 of Nm2e. In my opinion, it would be less confusing if the SpCas9 result squares were shifted to the right along with the spacer sequence, so that the corresponding identical nucleotides would be aligned.

-Fig. 2. b,c: It would be more informative to compare pairwise the corresponding WT and chimeric Nme2ABE8e variants, chimeric PID Nme2ABE8e-i7 and -i9 with WT PID Nme2ABE8e-i7 and -i9.

-Connected to the experiment: it would have been interesting to see, although it may be beyond the scope of the present work, how the PID swap affected the PAM preference and nuclease activity of the Nm2e nuclease.

- Lines 180-182: "These results indicate that Nme2Smu-ABEs can target and install precise edits at sites with N4CN PAMs, with a modest preference for N4CT PAMs for the target sites tested in our panel." I am not convinced that the observed higher activity is actually caused by the N4CT PAM preference. When testing a base editor on genomic targets, there are many variables that change from target to target and have a big impact on the editing efficiency: the not well known PAM preference of the nuclease (this is what we are testing), the target preference, the sequence specificity of the deaminase (in which N-A-N sequence context it works more efficiently), or where the maximally edited adenines fall in the editing window of each variant. Thus, when looking at only 3-3 targets per PAM, it is difficult to say with certainty which PAM the variants work more efficiently on. Therefore, to test a parameter on genomic targets, such as here the PAM preference of these variants, would require testing a very large number of targets per PAM. I think it is more appropriate to use reporter systems where target sequences can be freely specified so that, for example, the same targets can be examined in the same sequence environment and editing window position to determine PAM preference. With proper design, this can reduce the number of uncontrolled variables. We have also developed such a system (<https://www.nature.com/articles/s41467-021-26461-y>), which can be useful for deciding such questions. However, it may be that the authors, if they agree with me in raising the problem, just want to modify the incriminated sentence.

- Lines 150-151: "Like the domain-inserted Nme2-ABEs, the Nme2-CBEs exhibited insertion site-dependent shifts of their editing windows." I'm not sure how the editing window of the variants can be confidently inferred from the small number of targets and 'C' positions examined.

-Fig2c The title of the figure mentions 15 targets, but only 14 are shown.

-Fig. 3c shows the on- and off-target sequence and editing of the otSG2 site. Not only is the PAM changed at the off-target location (so neither of the variants should work on it if they have N4CN preference), but if I interpret the figure correctly, none of the on-target 'A' are present in the off-target sequence, so neither is the maximum editable. Thus, it is difficult to interpret the small edit detected by the -i1 variant.

The authors write "The editing efficiency at the maximally edited adenine for each target was plotted." This could be interpreted to mean that if a new "A" appeared in an off-target sequence, which was then plotted as the " maximally edited adenine" by one of the variants, then the off-target sequence was considered to be another target and that "A" was plotted. This might be worth clarifying!

In my opinion, it would be most informative to plot the desired on-target edit (which here could be the maximally edited adenine) and the sum of the off-target adenine edits for each variant. This might solve this problem.

Ervin Welker

Reviewer #2:

Remarks to the Author:

Bamidele and colleagues used rational design to engineer a Nme2Cas9 domain inlaid base editor. After initial profiling they then swapped the Nme2Cas9 PAM-interacting with that of SmuCas9, to broaden the targetable sites. This approach resulted in compact, high fidelity DNA editor.

Overall, the paper is well written and easy to follow. The results are well presented.

I have only a few minor suggestions for improvement:

- Line 307/308: I would reword to show the non-significant p value and state "albeit this was not

statistically significant ($p=$).” To replace the phrase: “... though with substantial efficiency variability that compromises statistical significance.”

- In Figure 4b & 4d - it may be clearer in the figures to not include the legend within the data frame of the figure.

Reviewer #3:

Remarks to the Author:

In this study, Nathan Bamidele and colleagues engineered Nme2Cas9 to get several Nme2Cas9 adenine base editors. The authors showed that domain-inlaid Nme2Cas9 variants exhibited shifted editing windows and increased activity in comparison to the N-terminally fused Nme2-ABE. They used these enhancements to correct two common mutations associated with Rett syndrome and validated domain-inlaid Nme2-ABEs for single-AAV delivery in vivo.

1. Overall, I think this is a very interesting to optimize Nme2-ABE by domain domain-inlaid method. To provide a more comprehensive view of editing windows and editing efficiency, the editing results of 15 endogenous target sites were analyzed by the Nme2-ABE8e effectors, all Nme2-ABE8e effectors exhibited average editing less than 30% as shown in Supplementary Fig. 1c. “The inlaid Nme2-ABE8e effectors showed comparable activity as Spy-ABE8e at six out of eight of the target sites (Fig. 1c-d).” The authors should avoid cherry picking, they should confirm this conclusion at many more (at least 20) sites and in at least two different cell lines.

2. The PAM scope of Nme2Cas9 has been expanded and several high-activity variants have been developed by Tony P.H., et al. “High-throughput continuous evolution of compact Cas9 variants targeting single-nucleotide-pyrimidine PAMs” Nature Biotechnology (2022). Variants eNme2-T.1 and eNme2-T.2 offer access to N4TN PAM, while eNme2-C and eNme2-C.NR show comparable or higher activity in a variety of human cell types and lower off-target activity at N4CN PAM sequences. The comparative data with the existing Nme2Cas9 variants should be presented. Does this manuscript contain fundamental results of sufficient novelty and significance to justify publication in Nature Communications?

3. Analysis of domain-inlaid Nme2-ABE8e specificity. “We selected four target sites for assessment, of which three had been validated as detectably edited off-target sites” the data for four target sites can’t support the conclusion.

4. In the Results, line 224-224, please check the use of the Supplementary Figure are correct, for the comparison of the off-target editing between the Spy-ABE8e and the Nme2ABE8e variants.

RESPONSE TO CRITIQUES

We are grateful to the referees for their detailed, thoughtful, and constructive comments and critiques, which we have diligently addressed. In particular, in response to several points raised by the reviewers, we have greatly expanded the numbers of guide/target/effector combinations analyzed in the manuscript through the use of high-throughput guide/target library experiments. Our efforts to provide the requested information, experiments, and analyses have greatly improved the paper, and we appreciate the opportunities for improvement that the reviews have provided. Specific points in the critiques (in black font), and our responses (in blue font), are provided below.

REVIEWER COMMENTS

Reviewer #1 (Remarks to the Author):

In their manuscript, Bamidele et al. developed new base editors (BEs) combining the nickase version of Nme2Cas9 with different deaminases. The work built on previous work by the group in which they created an Nme2Cas9ABE8e variant that can be incorporated into Adeno-associated virus due to its compact size. They sought to improve the low activity of the variant by creating inlaid variants in which the deaminase domain was inserted at 8 different sites, based on the structure of Nme2Cas9, in the hope of better targeting of the deaminase domain to the non-targeted DNA strand. The 8 variants were tested on a genome-integrated cherry reporter system and the results were confirmed on 13 endogenous targets, where they could take into account the potentially different editing windows of the variants. Of the 3 most active inlaid variants, i1 showed a PAM distal, whereas i7 and i8 showed a PAM proximal editing window. Compared to SpCas9ABE8e, -i1 best approximated the activity of SpCas9ABE8e on 8 targets among the inserted variants.

Using the same insertion positions -i1, -i7 and -i8, two cytosine deaminases were integrated into Nme2Cas9 and their activity was monitored on 3 endogenous targets demonstrating that Nme2Cas9 is a flexible scaffold for insertion of a variety of deaminase domains.

In order to increase the targeting range, the PID (PAM interacting domain) of SmuCas9 which requires only a single 'C' PAM was replaced by the PID of Nme2 and a change in PAM preference was detected on 14 targets (5 on NNNNCC, 9 on NNNNCD PAM). Replacing the PID enabled the inlaid variants to function on NNNNCN targets.

The authors then tested the guide-dependent specificity of the variants with mismatch-containing guides in the cherry reporter system and at 4 genomic off-target sites. The inlaid variants showed higher off-target activity than the N-terminal variants, consistent with their higher on-target activity. The leader-independent off-target of the -i1, -i1-TadA8e V106W and N-terminal variants was investigated on SaCas9-generated R-loops. The inlaid variant showed reduced off-target activity compared to the N-terminal, while the TadA8e V106W mutation in the context of the inlaid variant further reduced off-target editing.

The utility of the variants was demonstrated in a patient-derived fibroblast by converting the second most common Rett syndrome mutation to WT, located in a pyrimidine-rich region where the previously

developed base editors cannot be used. In addition, the activity of the -i1 inlaid variant was demonstrated in vivo by AAV delivery and in a mouse model.

This paper addresses an interesting problem by creating base editors that, due to their small size, can be packaged in AAV and applied to pyrimidine-rich regions. The manuscript is well structured, scientifically well written, the experiments are logical and fit for purpose, and the figures are descriptive. In general, the results of the experiments provide a good basis for the conclusions, and the interpretations are not exaggerated.

Thank you for your positive feedback!

I really have only minor comments:

-Supp. Fig. 2 and Supp. Fig. 3 are reversed, they are called the other way around in the text. In what follows I use the numbers assigned to the figures/figure captions. Addressed and corrected.

-The legend of Supp. Fig. 2b mentions "On-target....R-loop assay". This term is not used in the MS text. Fixed.

-In lane 155, the authors refer to Supplementary Fig. 2d. 'd' panel is not included in either Supp. Fig 2. nor Fig 3. Probably, it should be Supplementary Fig. 3c. Fixed.

-Supp. Fig. 3a would be more informative if the figure also showed the position of UGI. Fixed.

-Supp. Fig. 3b legend. Somehow it was inserted: "n=3 biological replicates per off-target R-loop" although there is no off-target experiment in the figure. Fixed.

-Supp. Fig. 3c: The positions at the bottom of the figure run A1-A24. I assume they should run C1-C24. Fixed.

-Supp. Fig. 3c legend: "Summary of Mean C-to-T editing at the three endogenous HEK293T genomic loci with Nme2-*evo*FERNY or Nme2- *rAPOBEC1* constructs." In contrast, the caption on the panel says that only the *rAPOBEC1* result is shown, without the Nme2-*evo*FERNY one. Fixed.

"Crossed out boxes denote no cytidine at the position within the target's tested." It appears that the positions shown here do not correspond to the target positions of 'C' shown in panel 'b', although the data are from the same experiments as I understood from the relevant text (lines 147-151). Correct, in Supplementary Figure 2b we only selected a subset of cytidines with the highest editing to show exemplary data for specific spacers. Some cytidines were at the extreme ends within the target site (e.g. position 1 or 24 of the protospacer with no detectable editing observed). In Supplementary Figure 2c, we plotted the average editing for all available cytidines at the target sites tested. The full panel of data can be found in "Supplementary Table 1_source data".

-Fig 1e: When plotting the Dual-PAM spacer sequences, the two sequences are misaligned, as if the PAM of SpCas9 falls on position 21-23 of Nm2e. In my opinion, it would be less confusing if the SpCas9 result squares were shifted to the right along with the spacer sequence, so that the corresponding identical nucleotides would be aligned. Thank you for this suggestion, which we have now addressed.

-Fig. 2. b,c: It would be more informative to compare pairwise the corresponding WT and chimeric Nme2ABE8e variants, chimeric PID Nme2ABE8e-i7 and -i9 with WT PID Nme2ABE8e-i7 and -i9. To address this and several other points raised by multiple reviewers, we have added data from an additional experiment to comprehensively compare the editing windows and activities of our Nme2-ABE variants, in addition to eNme2-C from David Liu's lab (PMID: 36076084). For this experiment, we used a paired guide-target library with ~200 members, which we integrated into the genome of HEK293T cells.

This allowed us to thoroughly sample windows and activities of a far larger number of target sites in a pooled format. The supporting data can be found in Fig. 2b-d and Supplementary Figs. 4 to 6. We believe that this addresses this request for pairwise comparisons. Additionally, we have the pairwise comparisons of the corresponding WT and chimeric Nme2-ABE8e variants in Figure 3c, although the comparisons there are for a smaller subset of target sites.

-Connected to the experiment: it would have been interesting to see, although it may be beyond the scope of the present work, how the PID swap affected the PAM preference and nuclease activity of the Nme2 nuclease. We agree, though to maintain the current manuscript's focus on domain-inlaid base editors, and to stay within a reasonable manuscript length, we are deferring that work to a follow-up manuscript that includes multiple strategies (including but not limited to PID swapping) to further improve activities and targeting ranges of Nme2-derived nuclease and base editors.

- Lines 180-182: "These results indicate that Nme2Smu-ABEs can target and install precise edits at sites with N4CN PAMs, with a modest preference for N4CT PAMs for the target sites tested in our panel." I am not convinced that the observed higher activity is actually caused by the N4CT PAM preference. When testing a base editor on genomic targets, there are many variables that change from target to target and have a big impact on the editing efficiency: the not well known PAM preference of the nuclease (this is what we are testing), the target preference, the sequence specificity of the deaminase (in which N-A-N sequence context it works more efficiently), or where the maximally edited adenines fall in the editing window of each variant. Thus, when looking at only 3-3 targets per PAM, it is difficult to say with certainty which PAM the variants work more efficiently on. Therefore, to test a parameter on genomic targets, such as here the PAM preference of these variants, would require testing a very large number of targets per PAM. I think it is more appropriate to use reporter systems where target sequences can be freely specified so that, for example, the same targets can be examined in the same sequence environment and editing window position to determine PAM preference. With proper design, this can reduce the number of uncontrolled variables. We have also developed such a system (<https://www.nature.com/articles/s41467-021-26461-y>), which can be useful for deciding such questions. However, it may be that the authors, if they agree with me in raising the problem, just want to modify the incriminated sentence. Thank you for raising this point, with which we now agree. We have removed the incriminating sentence in the revision.

- Lines 150-151: "Like the domain-inserted Nme2-ABEs, the Nme2-CBEs exhibited insertion site-dependent shifts of their editing windows." I'm not sure how the editing window of the variants can be confidently inferred from the small number of targets and 'C' positions examined. This is another excellent point. We agree that editing windows cannot be confidently inferred from such a small number of target sites, though we still do observe a shift in where the maximum editing occurs within a target, depending on the domain inlaid effector used. We have changed the wording of the statement from editing window (which can't be confidently inferred), to editing hotspots.

-Fig2c The title of the figure mentions 15 targets, but only 14 are shown. Noted and corrected, thank you.

-Fig. 3c shows the on- and off-target sequence and editing of the otSG2 site. Not only is the PAM changed at the off-target location (so neither of the variants should work on it if they have N4CN preference), but if I interpret the figure correctly, none of the on-target 'A' are present in the off-target sequence, so neither is the maximum editable. Thus, it is difficult to interpret the small edit detected by

the -i1 variant. We selected this off-target site as it had been shown to have ~40% editing for eNme2-C (mainly with N₄CN PAM preference) in the recently published paper (PMID: 36076084, where otSG2 is referred to as S-Site5-OT2 in their naming convention for their Supplementary Figure 20g). We did not include eNme2-C as a control for this experiment, so it is hard to determine if the site is a true off-target for Nme2Smu-ABEs. Though very few off-target sites tested for Nme2Cas9 related manuscripts show detectable editing, we decided to include it since it was an off-target site that demonstrated robust editing. Regarding the second comment about the maximum editable 'A', we respond in the next comment below.

The authors write "The editing efficiency at the maximally edited adenine for each target was plotted." This could be interpreted to mean that if a new "A" appeared in an off-target sequence, which was then plotted as the "maximally edited adenine" by one of the variants, then the off-target sequence was considered to be another target and that "A" was plotted. This might be worth clarifying! Thank you for noting this – we agree and have clarified this in the text as requested. We have been plotting an alternative 'A' in the off-target sequence if it 1) appeared in the off-target, or 2) the maximum editable 'A' changes between the On- or Off-target site or the inlaid variant.

In my opinion, it would be most informative to plot the desired on-target edit (which here could be the maximally edited adenine) and the sum of the off-target adenine edits for each variant. This might solve this problem.

Ervin Welker

Reviewer #2 (Remarks to the Author):

Bamidele and colleagues used rational design to engineer a Nme2Cas9 domain inlaid base editor. After initial profiling they then swapped the Nme2Cas9 PAM-interacting with that of SmuCas9, to broaden the targetable sites. This approach resulted in compact, high fidelity DNA editor.

Overall, the paper is well written and easy to follow. The results are well presented.

I have only a few minor suggestions for improvement:

- Line 307/308: I would reword to show the non-significant p value and state "albeit this was not statistically significant (p=)." To replace the phrase: "... though with substantial efficiency variability that compromises statistical significance." Agreed and corrected.

- In Figure 4b & 4d - it may be clearer in the figures to not include the legend within the data frame of the figure. Agreed and corrected.

Reviewer #3 (Remarks to the Author):

In this study, Nathan Bamidele and colleagues engineered Nme2Cas9 to get several Nme2Cas9 adenine base editors. The authors showed that domain-inlaid Nme2Cas9 variants exhibited shifted editing windows and increased activity in comparison to the N-terminally fused Nme2-ABE. They used these

enhancements to correct two common mutations associated with Rett syndrome and validated domain-inlaid Nme2-ABEs for single-AAV delivery in vivo.

1. Overall, I think this is a very interesting to optimize Nme2-ABE by domain domain-inlaid method. To provide a more comprehensive view of editing windows and editing efficiency, the editing results of 15 endogenous target sites were analyzed by the Nme2-ABE8e effectors, all Nme2-ABE8e effectors exhibited average editing less than 30% as shown in Supplementary Fig. 1c. “The inlaid Nme2-ABE8e effectors showed comparable activity as Spy-ABE8e at six out of eight of the target sites (Fig. 1c-d).” The authors should avoid cherry picking, they should confirm this conclusion at many more (at least 20) sites and in at least two different cell lines.

We agree that the selection of sites for comparison with Spy-ABE8e was limited, and we aimed to be appropriately careful with our conclusions, limiting statements to the experiment at hand. The idea of the experiment was to test whether the baseline activity increase of the domain-inlaid variants compared to the N-terminally fused Nme2-ABE8e remained true, with Spy-ABE8e as an additional benchmark. The sites were not cherry-picked but were selected as they had been previously validated sites with overlapping PAMs from our initial Nme2Cas9 nuclease paper, PMID: 30581144.

2. The PAM scope of Nme2Cas9 has been expanded and several high-activity variants have been developed by Tony P.H., et al. “High-throughput continuous evolution of compact Cas9 variants targeting single-nucleotide-pyrimidine PAMs” Nature Biotechnology (2022). Variants eNme2-T.1 and eNme2-T.2 offer access to N4TN PAM, while eNme2-C and eNme2-C.NR show comparable or higher activity in a variety of human cell types and lower off-target activity at N4CN PAM sequences. The comparative data with the existing Nme2Cas9 variants should be presented. Does this manuscript contain fundamental results of sufficient novelty and significance to justify publication in Nature Communications?

Thanks for raising this issue, given that we had comparative eNme2-C experiments already in progress when we submitted the original version of this manuscript. We now include comparative data for eNme2-C at a total of 15 endogenous target sites within human and mouse immortalized cell lines (HEK293T and Neuro2A respectively). In addition, as noted above, we have now conducted a paired guide-target library experiment for pairwise analyses of editing window and activity between the Nme2-ABE8e variants at ~200 target sites with a variety of PAMs. In these analyses, eNme2-C was included, and summary data can be found in Figure 2b and 2d. As is now shown, our engineered editors exceed the activity of eNme2-C in many instances. We note further that the adjustable editing windows are specific to our inlaid editors and are not offered by the reported eNme2-C platform. Our results also indicate that multiple avenues can lead to comparable independent platform technology outcomes, thereby adding considerable value to the field and providing additional basis for future platform improvement.

3. Analysis of domain-inlaid Nme2-ABE8e specificity. “We selected four target sites for assessment, of which three had been validated as detectably edited off-target sites” the data for four target sites can't support the conclusion.

We see the merit of this critique, but the limited panel of off-targets is due to the difficulty of finding *bona fide* off-target sites for Nme2Cas9 effectors, as reflected in numerous previously published papers using this platform. We included such sites when available, and we augmented those analyses with our mismatch specificity experiment with the reporter cell line (Figure 3b). Furthermore, we show that the enhanced editing activities of the inlaid effectors can increase off-target editing, though the efficiency

with which the off-target editing occurs differs between the inlaid editors. Finally, we tried to be appropriately cautious in our conclusions about specificity. In summary, we believe we have provided ample support for our limited specificity conclusions, within the confines of the overall shortage of validated off-target sites for this hyper-accurate editing platform.

4. In the Results, line 224-224, please check the use of the Supplementary Figure are correct, for the comparison of the off-target editing between the Spy-ABE8e and the Nme2ABE8e variants. Corrected, thank you.

Reviewers' Comments:

Reviewer #1:

Remarks to the Author:

The authors have adequately addressed all the issues raised. The generation and examination of the target library has greatly improved the manuscript. Looking at the data, it appears that the ~200 targets already reach a number of targets that can be used to obtain consistent results for defining the editing window, whereas the N4CC results obtained with just about 40 targets are still not a large enough number to obtain consistent results at all positions. In this respect, this work sets a sort of standard for how many target sequences should be used to define an editing window (at least about 200). This might be worth highlighting in a sentence or two in the results or discussion section.

I have a couple of minor issues:

- The way of tracking the changes in the introduction was not very helpful to me.
- Lines 164-167: Changing the PAM from N4CC to N4CN doesn't double but quadruple the number of available targets, does it?
- Lines 169-171: Figure 2a should be Supplementary Figure 3a instead of Figure 2a.
- The text mentions testing 6 N4CC and 12 N4CD endogenous targets, but the figure shows only 14 of the 18 targets, 4 N4CC and 9 N4CD.
- Given the relatively large number of such errors in the manuscript, I would encourage the authors to carefully double-check all citations and figure descriptions, which I have not done here...

Ervin Welker

Reviewer #3:

Remarks to the Author:

In this paper, Nathan Bamidele et al. constructed domain-inlaid Nme2-ABEs, these variants exhibited shifted editing windows and increased activity in comparison to the N-terminally fused Nme2-ABE, and the Chimeric Nme2Smu-ABEs enable recognition of N4CN PAMs and expanded the editing scope. Finally, the authors validated domain-inlaid Nme2-ABEs for single-AAV delivery in vivo. In general, the research is comprehensive with detailed data; however, there are several issues that need clarification.

1. The author constructed domain-inlaid Nme2-ABEs and achieved recognition of N4CN PAMs through the replacement of the PID domain. The authors used eNme2-C-ABE8e as a control, the performance of eNme2-C-ABE8e was notably lower in editing efficiency at 183 N4CN PAM sites depicted in Figure 2d, diverging from the reported performance in existing literature. It is essential to inquire about the criteria for selecting these 200 integrated sites and whether there was a deliberate data selection process. I recommend incorporating genomic target points from the article 'High-throughput Continuous Evolution of Compact Cas9 Variants Targeting Single-Nucleotide-Pyrimidine PAMs' to further substantiate that Nme2Smu-ABEs indeed outperform the previously reported eNme2-C-ABE8e in editing performance. This addition would enhance the value of the article and justify its publication.

2. In the portion of the detection of the domain-inlaid Nme2-ABE8e specificity and guide-independent off-target editing, eNme2-C-ABE8e should be considered as a control.

3. The guide-dependent off-target editing of domain-inlaid Nme2-ABE8es should be tested.

4. There is a lack of necessary literature citations in some places, such as the introduction line 47 to line 48.

RESPONSE TO CRITIQUES

We are again grateful to the referees for their helpful comments and critiques, which we have addressed in this re-revision. Specific points in the critiques (in black font), and our responses (in blue font) are provided below.

Reviewer #1 (Remarks to the Author):

The authors have adequately addressed all the issues raised. The generation and examination of the target library has greatly improved the manuscript. Looking at the data, it appears that the ~200 targets already reach a number of targets that can be used to obtain consistent results for defining the editing window, whereas the N4CC results obtained with just about 40 targets are still not a large enough number to obtain consistent results at all positions. In this respect, this work sets a sort of standard for how many target sequences should be used to define an editing window (at least about 200). This might be worth highlighting in a sentence or two in the results or discussion section.

Thank you for this suggestion. We considered adding such a sentence as the reviewer proposes but did not see a smooth and natural place to insert a separate reference to the relationship between library size and editing window. We also think that conclusive claims about this discussion would require a deeper empirical analysis of libraries of various sizes. Accordingly, we have chosen to defer this discussion to subsequent work.

I have a couple of minor issues:

-The way of tracking the changes in the introduction was not very helpful to me. We apologize for the fact that, for reasons we do not understand, Microsoft Word flagged certain entire paragraphs as changed, when only small sections of those paragraphs had in fact been edited. We made multiple attempts to fix this, as we discussed with the editor at the time of submission, but we did not succeed in making the program overcome this issue. We proceeded with resubmission nonetheless to avoid undue delays in re-review.

- Lines 164-167: Changing the PAM from N4CC to N4CN doesn't double but quadruple the number of available targets, does it?

Yes – thank you for highlighting this error, which we have now corrected.

- Lines 169-171: Figure 2a should be Supplementary Figure 3a instead of Figure 2a.

Thank you for this comment. Our intention here was to highlight Figure 2a (the predicted Nme2^{Smu}Cas9 homology model) rather than the data itself, but we were insufficiently clear about this. We now have a separate reference to Supplementary Figure 3a in the sentence in lines 171-173 to make this more explicit to the audience.

- The text mentions testing 6 N4CC and 12 N4CD endogenous targets, but the figure shows only 14 of the 18 targets, 4 N4CC and 9 N4CD.

Corrected, thank you.

- Given the relatively large number of such errors in the manuscript, I would encourage the authors to carefully double-check all citations and figure descriptions, which I have not done here...

Noted with apologies, thank you. We have double-checked as requested, and we believe that these errors have been eradicated in the re-revision.

Reviewer #3 (Remarks to the Author):

In this paper, Nathan Bamidele et al. constructed domain-inlaid Nme2-ABEs, these variants exhibited shifted editing windows and increased activity in comparison to the N-terminally fused Nme2-ABE and the Chimeric Nme2Smu-ABEs enable recognition of N4CN PAMs and expanded the editing scope. Finally, the authors validated domain-inlaid Nme2-ABEs for single-AAV delivery in vivo. In general, the research is comprehensive with detailed data; however, there are several issues that need clarification.

1. The author constructed domain-inlaid Nme2-ABEs and achieved recognition of N4CN PAMs through the replacement of the PID domain. The authors used eNme2-C-ABE8e as a control, the performance of eNme2-C-ABE8e was notably lower in editing efficiency at 183 N4CN PAM sites depicted in Figure 2d, diverging from the reported performance in existing literature. It is essential to inquire about the criteria for selecting these 200 integrated sites and whether there was a deliberate data selection process. I recommend incorporating genomic target points from the article 'High-throughput Continuous Evolution of Compact Cas9 Variants Targeting Single-Nucleotide-Pyrimidine PAMs' to further substantiate that Nme2Smu-ABEs indeed outperform the previously reported eNme2-C-ABE8e in editing performance. This addition would enhance the value of the article and justify its publication.

Thank you for this comment. eNme2-C-ABE8e was first described and validated in “High-throughput Continuous Evolution of Compact Cas9 Variants Targeting Single-Nucleotide-Pyrimidine PAMs” by Huang *et al.*, as cited in our manuscript. In their paper, Huang *et al.* compared eNme2-C-ABE8e to its unevolved variant (Nme2-ABE8e-nt) at a limited panel of target sites with the native N₄CC PAM. They observed an increase of ~1.3-fold activity on N₄CC PAMs for ~4 target sites in the bacterial ABE-PPA assay and a ~1.8-fold increase in activity for ~8 N₄CC PAM targets in HEK293T cells. For our library, we selected top-performing guides from a variety of Nme2Cas9 publications (as referenced), including 31 sites from the Huang *et al.* publication (see Supplementary Table 2, guide-target library, reference column), with a variety of N₄CC PAM targets. We have amended line 195-197 in the manuscript to better clarify target site selection criteria for the readers.

It is not clear to us what the reviewer is trying to imply when inquiring about a “deliberate data selection process,” but – in keeping with our groups’ practice with all our papers – all experimentally valid data in our editing analyses were described and used, without cherry-picking or sweeping subsets of results under the rug. For data analysis, the only target sites that were filtered were the small number that exhibited insufficient sequencing depth to support valid conclusions, but these sites were filtered within all samples to enable accurate comparisons, as described in the “Guide-target library editing and analysis” methods section (and in keeping with the established standards of the field).

We disagree that our analyses deviate significantly from the reported literature. Regrettably, when the reviewer states that “the performance of eNme2-C-ABE8e was notably lower in editing efficiency at 183 N₄CC PAM sites depicted in Figure 2d, diverging from the reported performance in existing literature,”

the precise basis for that statement is not specified. Absolute editing efficiencies cannot be compared directly when so many differences in experimental conditions exist [endogenous sites vs. library-based, the nature of the effector expression constructs (transposons with drug selection vs. transient plasmid transfection), the corresponding effector expression levels, etc. etc.]. Because we used matched expression constructs and parallel analyses for all effectors in our library-based experiments, relative editing values of different effectors provide a much more valid basis for efficiency comparisons. Our results are in fact highly consistent with previous reports in two crucial and internally consistent ways. First, the guide-target library assay captures the previously reported editing window and window center for eNme2-C-ABE8e observed in the Huang *et al.* publication (lines 208-211). Second, when we compare the mean activity of the window center for eNme2-C-ABE8e and Nme2-ABE8e-nt at 41 N₄CC PAM target sites, we observe a ~1.3-fold activity increase in activity for the former (Supplementary Figure 6a), in good agreement with the initial eNme2-C paper. At the time of writing our revision, no other comparisons had been made for Nme2^{Smu}-ABE editors and eNme2-C-ABE, and thus we are unaware of any other basis for implying that Figure 2d diverges from reported performance in the literature.

Another crucial point is that, when we prepared our revision, previous reports only compared eNme2-C-ABE8e (with its N-terminally fused deaminase domain) to the parental wild-type Nme2-ABE8e construct, also with an N-terminal fusion. The reviewer asks that we “further substantiate that Nme2^{Smu}-ABEs indeed outperform the previously reported eNme2-C-ABE8e in editing performance.” However, Figure 2d shows clearly that Nme2^{Smu}-ABE8e-nt does not outperform eNme2-C-ABE8e – the reviewer fails to note or acknowledge this, and we never claimed otherwise. The editing improvements over that of eNme2-C that we report here appear to come primarily the repositioned deaminase domain. The improved efficacy of the domain-inlaid variants is, of course, one of the central points of the manuscript.

We are currently developing and analyzing our next generation of Nme2^{Smu}-ABE and nuclease systems for a follow-up publication and are carrying out additional comparisons and validations of eNme2-C, eNme2-C.NR and Nme2^{Smu}-ABE derivatives that will be reported in that manuscript.

2. In the portion of the detection of the domain-inlaid Nme2-ABE8e specificity and guide-independent off-target editing, eNme2-C-ABE8e should be considered as a control.

Unlike with the guide-target library analyses, we did not yet have access to eNme2-C when we did our off-target analyses at endogenous sites, and we considered it then (and still consider it) unnecessary to go back and re-do all these analyses with an additional effector. The purpose of this assay was to provide an apples-to-apples assessment of how deaminase insertion altered fidelity in comparison to the N-terminal fused construct. Laborious re-tests with the additional evolved and mutated editor would not have been consistent with this purpose and would also have unduly delayed our ability to inform the field of the efficacy improvements we made by inlaying the deaminase domains.

As noted above, we continue to engineer and improve the Nme2Cas9 and Nme2^{Smu}Cas9 platforms for base editing and nuclease applications, and additional comparisons with eNme2-C and other emerging variants are ongoing. We are deferring those results for subsequent manuscripts.

3. The guide-dependent off-target editing of domain-inlaid Nme2-ABE8es should be tested.

We have already tested guide-dependent off-targeting editing for domain-inlaid Nme2-ABE8e's in two ways. First, in Figure 2c, we test guide-dependent off-targets at previously validated GUIDE-seq off-

targets at endogenous genomic loci within HEK293T cells. Second, in Figure 3b, we use a synthetic mismatch tiling assay in a reporter cell line. Thus, the tests that the reviewer requests have already been provided. Further analyses of on- vs. off-target edits at various sites of interest will be ongoing as our roster of sites grows, and we defer additional analyses to later publications.

4. There is a lack of necessary literature citations in some places, such as the introduction line 47 to line 48.

Thank you for pointing out this oversight. Citations have been added in the re-revision.